



# Fully Coupled High-Resolution Atmosphere-Ocean-Wave

# Simulations of Hurricane Henri (2021): Implications for Offshore

# Load Assessments

Chunyong Jung[1], Pengfei Xue[1,2,3], Chenfu Huang[2,3], William Pringle[1], Mrinal Biswas[4], Geeta Nain[1,2], and Jiali Wang[1]

[1]Environmental Science Division, Argonne National Laboratory, Lemont, IL 60439, USA
[2]Department of Civil and Environmental Engineering, Michigan Technological University, Houghton, MI 49931, USA
[3]Great Lakes Research Center, Michigan Technological University, Houghton, MI 49931, USA
[4]National Center for Atmospheric Research, Boulder, CO 80310, USA

*Correspondence to*: Chunyong Jung (cjung2@anl.gov); Pengfei Xue (pexue@mtu.edu)

**Abstract.** This study presents a fully coupled modelling system that integrates atmospheric, ocean, and wave models to simulate interactions during tropical cyclones and assess their implications for offshore infrastructure. The system is evaluated using Hurricane Henri (2021), chosen for its distinctive track along the U.S. northeast coast, an area with densely populated regions and offshore wind energy zones. The event is supported by extensive observations, including airborne Doppler radar, dropsondes, sea surface temperature, and ocean surface wave measurements. Three experiments with increasing complexity in atmosphere-ocean-wave coupled processes are conducted to examine their impact on storm intensity and development. Compared to atmospheric-only and atmosphere-ocean coupled simulations, the fully coupled model reduces intensity overestimations and improves the wind structure from near the surface to the upper troposphere. These improvements are due to realistic representation of complex feedback loops between the atmosphere, ocean, and waves. Wave-induced cooling of sea surface temperatures and reduced surface enthalpy flux mitigate intensity overestimation. Additionally, wave-driven surface roughness, reflected in realistic surface roughness length and drag coefficients, enhances the radial and vertical profiles of hurricane boundary layer winds. The fully coupled simulation shows promising potential for assessing risks to offshore infrastructure, featuring a more stable atmospheric boundary layer, weaker surface roughness, and lower turbulent kinetic energy. These factors allow wind veer to persist and align more closely with observations. The system also captures wind-wave misalignment, emphasizing the importance of incorporating ocean and wave components for accurate risk assessments in offshore infrastructure, such as wind turbine operations.





## 1 Introduction

Tropical cyclones (TCs) pose a serious threat to society, bringing destructive winds, large waves, storm surges, heavy rainfall, and compound flooding. Over the past four decades (1980-2019), TCs have led U.S. disaster records, causing the highest economic losses ($945.9 billion) and fatalities (6,502) among all natural hazards (Smith, 2020). Climate warming further complicates TC behaviour, adding uncertainties: while TC frequency may remain stable or decline, TC intensity is likely to increase in many coastal areas, exacerbating coastal hazards and offshore energy harvesting (Knutson et al., 2010, 2019, 2020; Walsh et al., 2015, 2016; Wang et al., 2024a,b). Consequently, accurate TC prediction has become increasingly essential for effective risk analysis, disaster prevention, and infrastructure design.

Forecasting TC tracks has improved substantially over the past few decades, while predicting TC intensity remains challenging and has shown only limited progress (DeMaria et al., 2014; Rappaport et al., 2009; Yamaguchi et al., 2017; Zao et al., 2022). Previous studies suggested that insufficient consideration of the physical processes and their subsequent heat and momentum exchanges at the atmosphere-ocean interface might be responsible for the slower progress in improving TC intensity forecasts compared to track forecasts over the decades (DeMaria et al., 2007; Zhao et al., 2017, 2022). For example, Mogensen et al. (2017) and Wei et al. (2017) suggest that sea surface temperature (SST) cooling feedback process in TC modelling is a major factor contributing to bias in TC intensity forecasts. The storm extracts energy from the ocean through exchanges of heat, moisture, and momentum fluxes at the atmosphere and ocean interface. TC-driven high winds and waves enhance turbulent mixing in the upper ocean, resulting in SST cooling—often called "cold wakes"—along the storm's path. This process acts as negative feedback on storm energetics by reducing the surface enthalpy flux supply (Bender et al., 1993; Cavaleri et al., 2012; Cione and Uhlhorn, 2003; Fan et al., 2010; Schade and Emanuel, 1999). For instance, Zhu and Zhang (2006) found that cooling in SST, averaging -1.3 °C near the TC centre and along its track, statistically weakens hurricane intensity by about 25 hPa. This feedback mechanism between the atmosphere and ocean is primarily driven by vertical mixing of cooler waters from beneath the seasonal thermocline, induced by TC-generated strong waves, large upper-ocean shears, and upwelling from divergent ocean currents (Emanuel, 1986; Schade and Emanuel, 1999; Wu et al., 2016). The reduced surface heat exchange then weakens the moist enthalpy flux from the ocean to the atmosphere, thereby diminishing TC intensity. Therefore, to realistically capture TC-induced SST cooling, the atmospheric forecast model must be coupled with a three-dimensional ocean model (e.g., Yablonsky and Ginis, 2009).

Another key ocean element in TC evolution involves ocean surface waves, which significantly affect both atmospheric and ocean dynamics. As for the impact of the waves on atmospheric dynamics, on the one hand, ocean surface waves characterize surface roughness, influencing the structure of atmospheric and marine boundary layers. This, in turn, affects atmosphere-ocean momentum and heat exchanges, upper-ocean mixing, sea spray production, and albedo (Cavaleri et al., 2012). Liu et al. (2011) found that their satellite-based latent heat flux data (XseaFlux) performed significantly better in capturing TC-





associated latent heat flux by incorporating sea surface wave features such as wave breaking, wave orbital motion (non-breaking waves), and sea spray. Chen et al. (2007, 2013) emphasized the importance of wind-wave feedback under extreme conditions through directional wind-wave coupling, which enhances simulations of hurricane-induced surface winds and hurricane structure. Similarly, Zhao et al. (2017, 2022) found that wave-induced processes, including mixing and sea spray production, reduce biases in TC intensity forecasts, underscoring the essential role of ocean surface waves in atmosphere-ocean enthalpy flux exchanges and TC evolution. On the other hand, breaking waves also generate sea spray, which enhances atmosphere-ocean heat and moisture flux exchanges under tropical cyclone (TC) conditions, potentially intensifying TCs (Perrie et al., 2004, 2005; Richter and Stern, 2014; Zhao et al., 2022). However, Prakash et al. (2019) found that incorporating sea spray has only a marginal effect on storm intensity, suggesting that wave impacts on surface roughness may play a more significant role in the coupling process. This relative importance, however, requires further investigation with additional storm cases, as will be discussed in Section 6.

As for the impact of the waves on ocean dynamics, wave-induced mixing—driven by both breaking and non-breaking waves plays a key role in modulating SST, which is critical for TC development and intensity (Babanin, 2006; Qiao et al., 2004; Sullivan and McWilliams, 2010). Breaking waves create surface-level turbulence, but kinetic energy dissipates quickly with depth, limiting their influence on SST and surface heat fluxes (Craig and Banner, 1994; Zhang et al., 2007). In contrast, non-breaking waves penetrate deeper, significantly affecting SST and mixed layer depth, both of which are essential for TC intensity modification (Babanin et al., 2009; Qiao et al., 2004; Zhang et al., 2022). Additionally, waves influence the bottom boundary condition for TCs through wave-current interactions, affecting sea surface currents (Lane et al., 2007; Mellor, 2016; Olabarrieta et al., 2010; Smith, 2006) and SST through mechanisms such as radiation stress, Stokes drift, and vertical mixing (Wang et al., 2024b). Coupled wind-wave-ocean models underscore the significant impact of wind-wave-current interactions on atmosphere-ocean momentum flux and ocean responses in TCs (Fan et al., 2009).

Grid spacing is another critical factor being continually addressed as computational resources increase over time. Resolving the inner core of a TC with a grid spacing of 4 km or less has enabled the explicit representation of deep convection, leading to a more accurate depiction of TC structure in the atmospheric component (e.g., Gentry and Lackmann, 2010). In an atmosphere-ocean coupling framework, Tsartsali et al. (2022) emphasized that optimal results require higher resolutions in both ocean and atmospheric models, specifically at least eddy-permitting (~25 km) and better eddy-resolving (~8 km) ocean resolution, along with comparable atmospheric resolution, for reliable atmosphere-ocean coupling along the Gulf Stream. Additionally, Zhang et al. (2023) employed the Community Earth System Model at very high spatial resolutions (up to 3 km for the ocean and 5 km for the atmosphere) to capture major weather and climate extremes, highlighting the importance of convection-permitting resolution and sub-mesoscale ocean eddies in modelling TC dynamics and eddy-mean flow interactions.



This study introduces a newly developed atmosphere-ocean-wave coupled modelling system that integrates a regional
atmospheric climate model with ocean and surface wave models, both of which operate on a high-resolution unstructured
mesh. This framework, while sharing similarities with the Coupled Ocean Atmosphere Wave and Sediment Transport
(COAWST, Warner et al., 2010) model in its coupled components, distinguishes itself through its ability to provide regional
refinement over areas of interest (e.g., offshore wind farms). Its capability to generate ultra-high spatial resolution for the ocean
mesh allows for more detailed and localized information, enhancing its applicability for site-specific analyses. In addition, we
incorporate the impacts of non-breaking waves into the coupling system to enable more realistic interactions between the
atmosphere, ocean, and waves. Utilizing this fully coupled system at a very high resolution (3 km for the atmosphere and 3
km for the ocean and wave models near the U.S. Northeast Coast), we investigate the effects of atmosphere-ocean-wave three-
way feedback on tropical cyclone (TC) development and demonstrate its relevance in assessing potential TC-induced risks for
offshore wind infrastructure. The model presented here provides a more realistic depiction of the complex interactions between
the ocean, waves, and atmosphere compared to many existing statistical-parametric models (e.g., Arthur, 2021; Chen et al.,
2024; Roldán et al., 2023) and idealized TC models (e.g., Sanchez Gomez et al., 2023), which often analyse wind and wave
interactions separately or exclude them altogether. These limitations may result in incomplete risk assessments, as wave
dynamics play a substantial role in TC behaviour and evolution, as previously discussed.
The development of the model, including detailed information on each model component and the coupler, is described in
Section 2. Section 3 describes the experimental design and data used for model validation using Hurricane Henri (2021) as a
working example. In Section 4, we present results and analysis. Implications for potential risks to offshore wind energy is
discussed in Section 5, followed by the summary and discussions in Section 6.




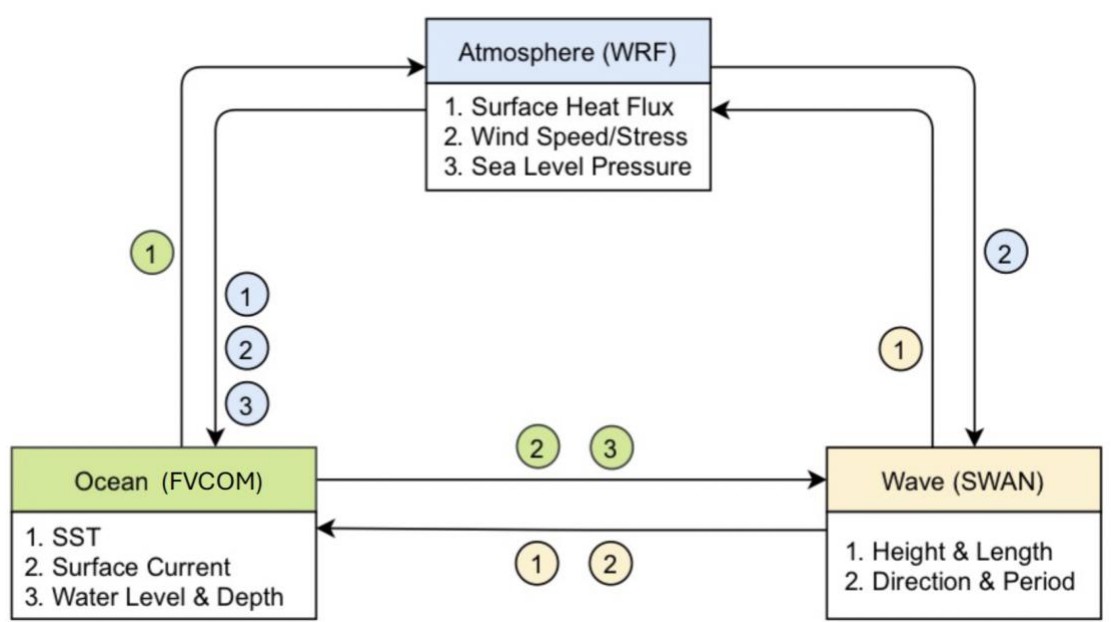

**Figure 1. Schematic of coupled atmosphere-ocean-wave system and modelling used in this study.**

## 2 Model Description

The coupled atmosphere-ocean-wave modelling system consists of three components: the Weather Research and Forecasting (WRF) model for atmospheric processes (WRF V4.5.1; Skamarock et al., 2019), the Finite Volume Community Ocean Model (FVCOM V4.3.1; Chen et al., 2003, 2013) for ocean circulations, the third-generation Simulating WAves Nearshore (SWAN) model for wave dynamics (Booij et al., 1999), and a coupler to exchange data fields (Fig. 1). Hereinafter, we refer to the coupled WRF-FVCOM-SWAN model as C-WFS. In the three-way coupled framework of C-WFS, the model components are executed in parallel, exchanging information through the Ocean Atmosphere Sea Ice Soil3 (OASIS3)-Model Coupling Toolkit (MCT) coupler (Craig et al., 2017). We describe each component, improvements made to them, and the approach to coupling in Sections 2.1-2.2.

### 2.1 Model Components

WRF is a nonhydrostatic, quasi-compressible atmospheric model with boundary layer physics schemes and a variety of physical parameterizations of sub-grid scale processes for predicting meso- and macroscales of motion. WRF has been extensively used for operational forecasts as well as for realistic and idealized research experiments. We have modified the WRF code to enable the wave slope-based sea surface roughness formulation from Taylor and Yelland (2001) in several





surface schemes (MYNN[Nakanishi and Niino, 2009; Olson et al., 2019], and the original and revised MM5 [Dyer and Hicks, 1970; Jimenez et al., 2012; Paulson, 1970; Webb, 1970]):

$$Z_0 = 1200 H_s \left(\frac{H_s}{L_p}\right)^{4.5} + \frac{0.11 v}{u_*} \qquad Z_0 \leq 0.00285 \qquad (1)$$

where $Z_0$ is the surface roughness length, $H_s$ is the significant wave height, $L_p$ is the wavelength at the peak of spectrum, $v$ is kinematic viscosity, and $u_*$ is the friction velocity. Other wave-based formulations (e.g., Drennan et al., 2003) are also available in C-WFS but in our testing we found that the capped Taylor and Yelland (2001) method gave the best performance for our case study.

The ocean model component, FVCOM, is a prognostic, free-surface, 3-D primitive equation coastal ocean circulation model that is numerically solved over an unstructured triangular grid using the finite-volume method. Version 4.3.1 of FVCOM is used in this study, allowing ocean hydrodynamic conditions to interact freely with atmospheric conditions throughout the simulation period. We modified the FVCOM code to incorporate vertical mixing effects induced by non-breaking waves. Non-breaking wave induced mixing is added to the turbulence eddy diffusivity $B_v$ included in the ocean model and is expressed as (Ghantous and Babanin, 2014a,b; Aijaz et al. 2017):

$$B_v = \alpha A^3 \kappa \sigma e^{3\kappa z} \qquad (2)$$

where $\alpha = 0.1$, A = wave amplitude ($H_s/2$), $\kappa$ = wave number ($2\pi/L$), $\sigma$ = peak wave frequency ($1/T_p$), z is water depth.

The wave model component, SWAN v41.01, is a third-generation spectral wave model developed at Delft University of Technology that computes random, short-crested wind-generated waves in coastal regions and inland waters (http://swanmodel.sourceforge.net/). It solves the evolution equation of wave action density in space time, frequency and wave direction dimensions (Pringle and Kotamarthi, 2021). Various wave energy sources and sinks are modelled, including wave generation by wind, wave decay due to whitecapping, bottom friction, depth-induced wave breaking, and energy redistribution through nonlinear wind-wave interactions.

**2.2 Coupler and Coupling**

OASIS3-MCT is a parallelized coupler that enables the simultaneous coupling of 2-D and 3-D fields. Figure 1 provides a schematic illustration of the C-WFS, detailing the quantities exchanged within the coupling framework. The friction velocity, surface winds, sea level pressure, latent and sensible heat fluxes, and shortwave and longwave radiation fluxes predicted by WRF are transferred to FVCOM as surface forcing, while FVCOM provides sea surface temperature (SST) to WRF as over-ocean boundary conditions. WRF supplies wind fields to drive SWAN for wave simulation, while SWAN provides significant wave height and wavelength at the peak of the spectrum to WRF, which uses them to calculate sea surface roughness based on equation (1). The wave fields are used by FVCOM to compute radiation stress gradients, enabling wave-driven flows,





Stokes velocities for mass flux transport, wave-enhanced bottom stresses, and non-breaking wave-induced mixing. Breaking
wave induced mixing is incorporated as a part of radiation stress gradients. Additionally, FVCOM provides sea surface currents
to SWAN, allowing for the inclusion of Doppler effects from background currents on surface waves. This integration enables
SWAN to better account for how ocean current movement affects wave behaviour, resulting in more accurate wave predictions.

## 3 Application of C-WFS Modelling System

This section describes the C-WFS setup used to simulate Hurricane Henri (2021). Henri reached Category 1 on the Saffir-
Simpson scale and made landfall in Rhode Island, U.S. on 22nd August 2021. Despite its relatively weak intensity, the storm
brought very heavy rainfall over the Northeastern U.S., including New England, causing widespread flooding and power
outages in the densely populated regions, such as New York and Boston. Moreover, Henri is one of the recent TCs to pass
through the offshore leased wind energy area in the U.S. northeast continental shelf. During this hurricane, comprehensive
observation datasets were collected, including airborne measurements such as doppler radars and dropsondes that reached the
eyewall and core. These conditions and datasets allow for direct comparisons between the modelled and observed TC
structures, providing insights into model performance and the coupling effects due to atmosphere-ocean-wave interactions.

### 3.1 Experimental Design and Configuration

To explore the integrated effects of ocean and ocean surface wave related physical processes on TC simulations, a set of three
model simulations is performed. WRF standalone simulation is named as experiment 'A', in which the event is modelled using
WRF alone with prescribed SST at 6-hour intervals. In experiment 'AO,' WRF is coupled with FVCOM, enabling variable
exchange as shown in Fig. 1, but without considering ocean surface wave-related physical processes. Experiment 'AOW' is a
multi-way fully coupled experiment, in which WRF, FVCOM, and SWAN exchange variables with each other every hour
through the OASIS3-MCT Coupler to allow direct and indirect atmosphere-ocean-wave interactions, as discussed in Section
185  2.
All simulations are initialized at 18:00 UTC on August 19, 2021, within a domain encompassing the western North Atlantic
Ocean. The atmospheric domain features a horizontal resolution of 3 km (Fig. 2a). The ocean domain, which covers a
substantial portion of the WRF ocean domain, employs an unstructured triangular grid with resolutions ranging from 3 km
near the coast to 9 km in the open ocean, effectively resolving the complex coastline of the U.S. Northeast Coast (Fig. 2b).
Initial and boundary conditions for the atmosphere model are obtained from the 6-hourly 0.25° NCEP (National Centers for
Environmental Prediction) Global Forecast System (GFS; NCEP, 2015) data. Note that the prescribed SST for experiment 'A'
is provided by GFS at 6-hourly intervals. The atmosphere is represented by 46 stretched vertical levels topped at 50 hPa with
12 layers below 100 metres. The physics selected for this study include the WRF single-moment 6-class microphysics scheme
(WSM6; Hong and Lim, 2006), the Rapid Radiative Transfer Model for GCMs longwave and shortwave schemes (Iacono et
al., 2008), the Yonsei University PBL (Hong et al., 2006), and the Eta similarity surface layer scheme which is based on the



revised MM5 Monin-Obukhov scheme (Jimenez et al., 2012). The land surface processes are modelled by the Noah (Chen and
Dudhia, 2001). No cumulus parameterizations are used in our WRF setup, as previous studies have demonstrated that a
resolution of 4 km or less is adequately convection-permitting in WRF for simulating extreme events (Akinsanola et al., 2024;
Kouadio et al., 2020; Qing and Wang, 2021; Sun et al., 2016).
Initial conditions for the ocean model fields of currents, water level, salinity, and temperature and boundary conditions for
currents, salinity, and temperature are derived from the $(1/12)°×(1/12)°$ resolution HYbrid Coordinate Ocean Model
(HYCOM; Cummings and Smedstad, 2014) analysis data (http://hycom.org/dataserver/) simulations. The ocean domain
employs varying horizontal resolution of ~9 km in the open ocean down to ~3 km over the continental shelf in the area of
interest (Fig. 2b). The ocean is represented vertically with 40 sigma layers, enabling the model to accurately reflect the abrupt
changes in coastal bathymetry. Vertical mixing processes are simulated using the Mellor–Yamada level-2.5 (MY25) turbulence
closure model (Mellor and Yamada, 1982), and horizontal diffusivity is computed using the Smagorinsky numerical
formulation (Smagorinsky, 1963).
For this study, the wave model domain covers the same geographic extent as the FVCOM domain with approximately 12 km
horizontal resolution. The wave spectrum is discretized into 36 directional bins and 24 frequency bins on the interval [0.04,1]
Hz. We use Komen et al. (1984) wave growth and whitecapping physics, Madsen et al. (1988) bottom friction, and a constant
depth-limiting wave breaker index, all with their default parameters. Lateral boundary conditions for swell are not applied due
to their insignificance at the eastern boundary.
All experiments involved a 102-hour integration, initialized from the same conditions at 18:00 UTC on August 19, 2021.
Following initialization, the simulations evolved freely throughout the entire 102-hour hindcast period without any technical
interventions. While nudging techniques, such as spectral nudging of variables such as wind, air temperature, and geopotential
height, are valuable for improving modelled tracks, they were intentionally not applied in this study. This decision reflects the
focus on exploring the impacts of multi-factor coupling between the atmosphere, ocean, and waves on tropical cyclone (TC)
characteristics. Applying nudging could complicate efforts to isolate the specific coupling effects of ocean and wave processes
on TC behaviour.
Several additional simulations were conducted using different planetary boundary layer and microphysics parameterizations,
as well as various forcing data (e.g., ERA5 reanalysis data, the fifth generation of ECMWF atmospheric reanalyses of the
global climate). The results consistently showed that the overall conclusions of this study remained unchanged, demonstrating
the robustness of the findings and their low sensitivity to these configuration choices.

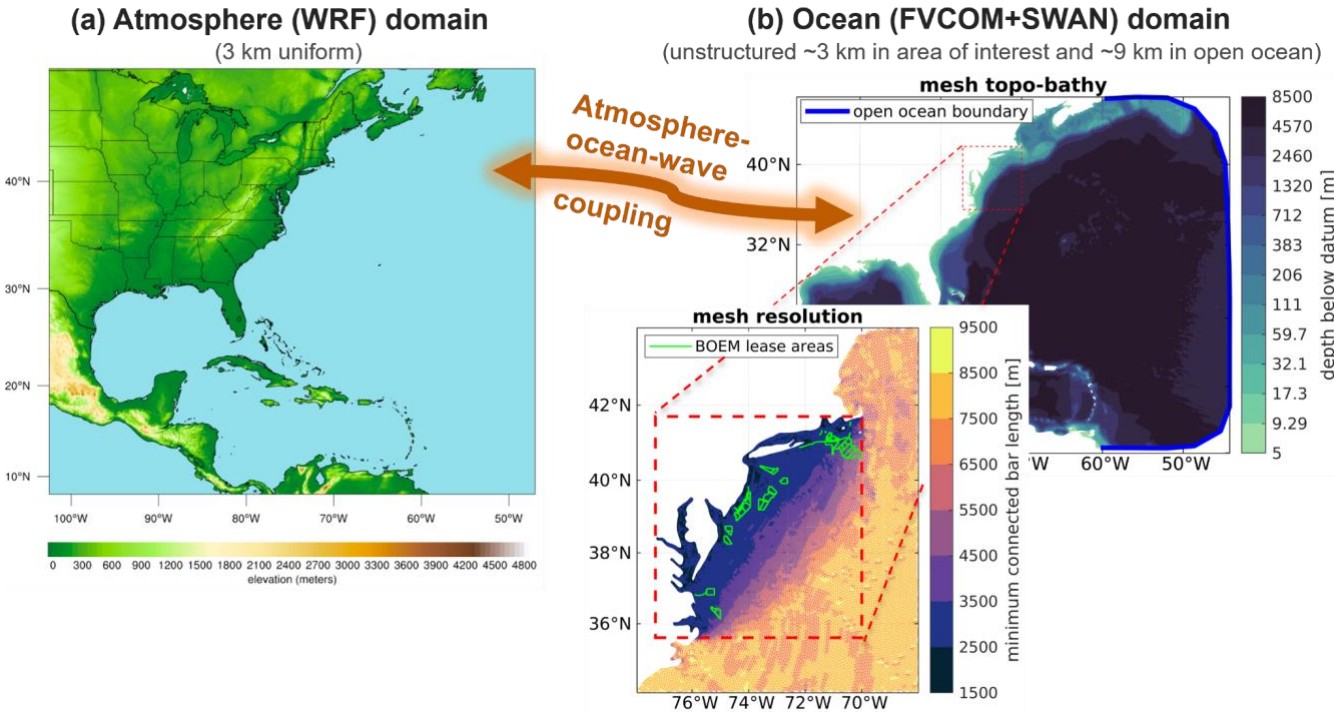

**Figure 2. (a) WRF model domain with terrain height elevation, and (b) FVCOM and SWAN domain with bathymetric depths and a zoom-in to the refined mesh grid along the northern U.S. East Coast and BOEM offshore lease areas.**

**3.2 Method and Data**

The model results are evaluated against observations using multiple datasets, including International Best Track Archive for Climate Stewardship (IBTrACS; Knapp et al., 2010) and airborne observations. IBTrACS is the most complete global collection of TCs, providing TC best position, minimum sea level pressure (SLP), maximum sustained wind speeds, and translation speed at mostly 6-hourly intervals. The airborne observations include the Tropical Cyclone Radar Archive of Doppler Analyses with Recentering (TC-RADAR) dataset (Fischer et al., 2022) and dropsondes from aircraft provided by the National Oceanic Atmospheric Administration's (NOAA) Hurricane Research Division (HRD). TC-RADAR is a comprehensive database of airborne observations of TCs, featuring data from the X-band tail Doppler radar on NOAA's WP-3D aircraft. This radar scans in both front and back directions, enabling detailed 3-D analyses of the inner-core structure of TCs. Typically, each mission includes 3-4 passes through the centre of the storm. For each central pass, an analysis is created using a technique called "recentring," which provides analyses on storm-cantered grids. We utilize storm-centred coordinates for our simulations, aligning the grids of TC-RADAR to enable direct comparison of the 3-D storm structure between the simulations and TC-RADAR. To align with the storm-centred grids of the TC-RADAR analyses, a 300 km × 300 km grid box is centered on the grid cell with the minimum SLP in each dataset. To provide seamless observations from the surface up to





0.5 km—the range not covered by TC-RADAR—we use dropsonde data as well. In this study, the model generates tracks and
translational speeds that differ slightly from the observations (Fig. 3). Therefore, the positions of the dropsondes are adjusted
relative to the storm centre rather than using their actual deployment locations. Here, we selected the seven dropsonde
observations shown as blue and colour dots in Fig. 3d for the assessment because they were deployed from a single flight
across the storm centre. This flight spanned from the eastern to the western edge within 50 minutes, just 12 hours before the
storm reached its peak intensity.
Modelled ocean surface waves are compared with observations from two National Data Buoy Center (NDBC; NDBC, 2008)
buoys, 41001 and 41002, located on the left of the storm track on the continental slope. While there are more buoy locations,
our focus is on the variation of storm-induced winds and waves along Henri's track. We exclude stations near the U.S Northeast
Coast due the models' track bias after 22$^{nd}$ August (more discussion in Section 4). The buoy data provides surface wind and
wave information, including surface wind speed, significant wave height, and peak wave period and direction. In addition to
in-situ NDBC buoy measurements, we compiled a series of daily SST data from the Operational Sea Surface Temperature and
Ice Analysis (OSTIA; Good et al., 2020) at $0.05° \times 0.05°$ resolution to determine the pre- and post-storm environment as well
as the difference between them.
The radius of maximum wind (RMW) defines the location of the maximum winds in a TC and is critical to understanding
intensity change as well as hazard impacts. In this study, we azimuthally average the vertical profiles of the seven dropsondes
and the simulations of wind speed relative to RMW to define the areas within and beyond the eyewall, allowing for a detailed
comparison of the storm's inner- and outer-core regions.










**Figure 3.** Comparison of the simulated (a) track, (b) minimum sea-level pressure (SLP), and (c) maximum surface wind speed of Hurricane Henri with the best track during the period from 18 UTC on 19 August to 00 UTC on 24th August 2021. Black lines represent values derived from IBTrACS observations. Green lines indicate the experiment 'A,' red lines depict the experiment 'AO,' and blue lines show the experiment 'AOW.' Figure (d) describes IBTrACS best track (grey line and dots) and dropsonde deployed positions (black and colour dots) during Hurricane Henri. Seven dropsondes (blue and colour dots) along the blue line are selected to assess model performance.

**4 Model Validation**

**4.1 Track and Intensity, and Storm Structure**

**4.1.1 Track and Intensity**

Figure 3 presents the tracks, SLP minima, and surface wind speed maxima derived from the three simulations alongside IBTrACS. The results indicate that variations in Henri's tracks across the three experiments are minimal (Fig. 3), consistent with previous findings suggesting that TC tracks are predominantly controlled by large-scale atmospheric circulation processes, rather than by atmosphere-ocean interactions at the temporal and spatial scales resolved in these models (e.g., Zambon et al., 2014). The root-mean square error (RMSE, Table 1) of position indicates all three simulations have similar track errors, with values of 123.7 km for 'A', 119.4 km for 'AO', and 126.1 km for 'AOW.' The relatively high error values are mainly due to significant deviations from the observed track after 00 UTC on 22nd August. These deviations are likely linked to biases in midlatitude upper-level atmospheric wave patterns, such as troughs and ridges, and their interactions with the storm, as the storms are deeply embedded in the baroclinic zone. Preliminary tests show that applying zonal and meridional nudging to winds, geopotential heights, and air temperature above the boundary layer can effectively improve track accuracy. However, as previously discussed, the primary objective of this study is to explore atmosphere-ocean-wave interactions in simulating the evolution and development of Hurricane Henri. Consequently, all results presented in the following sections are derived from simulations conducted without the use of any nudging techniques.

In terms of minimum SLP for the simulation of Henri's intensity (Fig. 3b), noticeable differences between the modelled storms begin to emerge 12 hours after the simulation starts. While all three simulated storms show an overestimation throughout nearly the entire lifecycle of the storm, especially when they reach their peak at 12 UTC on 22nd August, the magnitude of this overestimation is reduced in 'AO' and 'AOW' experiments compared to 'A.' In 'AOW', the overestimation of minimum SLP is delayed until 00 UTC on 22nd August. It then reaches the weakest minimum SLP value, resulting in the lowest RMSE in minimum SLP among the three simulations (Table 1). These temporal trends also apply to the maximum surface wind speed (Fig. 3c and Table 1), demonstrating a reduction in overestimation of maximum surface wind speed in both 'AO' and 'AOW' experiments compared to 'A.' Between the experiments 'AO' and 'AOW,' while 'AOW' generally exhibits weaker wind speeds compared to 'AO,' it becomes stronger as the storm approaches and reaches its peak intensity, in contrast to the findings for minimum SLP. The potential physical processes underlying this discrepancy are discussed in Section 4.3. It is important to note that the maximum wind speeds recorded in the three simulations represent the peak values at a given time on an hourly





basis, which do not fully capture the sustained wind conditions or the broader characteristics of the simulated storms. On the
other hand, the observed maximum wind speeds are recorded at a single location and time step, which may not adequately
reflect the full extent of potential damage caused by high wind conditions. Therefore, a more thorough evaluation of storm
structure, both near the surface and across multiple altitudes, is necessary to provide a more complete assessment of the model's
performance.

**TABLE 1. Root mean square error (RMSE) for each simulation in terms of minimum sea level pressure (hPa), maximum surface wind speed (m s$^{-1}$), and cyclone track (km).**

| Experiment | Minimum sea level pressure (hPa) | Maximum surface wind speed (m/s) | Cyclone track (km) |
|---|---|---|---|
| A | 9.4 | 10.2 | 123.7 |
| AO | 7.9 | 8.7 | 119.4 |
| AOW | 6.4 | 8.3 | 126.1 |

**4.1.2 Storm Structure**

Here we examine the model's performance in simulating the three-dimensional storm structure of Henri. Figures 4a-d show
the airborne Doppler radar-observed (TC-RADAR) wind speeds at 1-km level and vertical profiles in Henri at 00 UTC on 21$^{st}$
and 22$^{nd}$ August 2021 along the black lines shown in Figs. 4a-b. On 21$^{st}$ August, the storm shows a distinct asymmetric
distribution of strong winds, primarily concentrated on the right side—a characteristic of a tropical storm transitioning into a
hurricane (Fig. 4a). This asymmetry is largely due to the combination of the storm's poleward movement and its cyclonic
circulation. As Henri moves northward during this period, the winds on the right side effectively combine with the storm's
forward motion, leading to higher wind speeds. The vertical cross-section of wind speed along the line from A to B, shown in
Fig. 4a, offers a different perspective by contrasting the winds between the northern and southern areas. It shows that wind
speeds exceeding 20 m s$^{-1}$ are mostly concentrated below 4-km level in the southern part, while strong winds extend up to 8-
km level in the northern part at this time (Fig. 4c). At 00 UTC on 22$^{nd}$ August — 12 hours before reaching peak minimum SLP
intensity, Henri exhibited a compact and nearly closed distribution of strong winds exceeding 24 m s$^{-1}$ along the eyewall. This
demonstrates a more organized, symmetric appearance, while relatively weaker wind zones remain on the left side (Figs. 4b,d).
The vertical cross-section clearly illustrates the structural changes that Henri underwent; it reveals a distinct calm wind zone
within the eyewall, extending up to 9-km level. Areas of strong wind speeds exceeding 24 m s$^{-1}$ are relatively evenly distributed
relative to the centre, with the strongest winds located on the right side. The corresponding simulated vertical profiles and
horizontal distributions of wind speed at 1-km level at 00 UTC 21$^{st}$ and 22$^{nd}$ August 2021 are shown in Figs. 4e-p.

All three simulated storms reasonably capture the structural changes that Henri underwent, including the transition in wind
distribution from a wide, open, asymmetric pattern with strong wind zones on the right side observed at 00 UTC on 21$^{st}$ in TC-
RADAR to a more compact, closed, symmetric structure as it intensifies observed at 00 UTC on 22$^{nd}$ August in TC-RADAR.





However, the simulated storms noticeably overestimate intensity horizontally and vertically, especially the experiments 'A'
and 'AO.' The fully coupled simulation 'AOW' notably mitigates the overestimation with better radial wind profiles at the 1-
km level along the line from A to B for both times (Fig. A1), and higher Pearson correlation coefficients (r) of 0.95 and 0.72,
respectively—the highest correlations among the three simulations at both times. For a more comprehensive examination, we
assess the wind distribution using probability density function (PDF) considering all available observation grid cells
horizontally and vertically within the 300 km x 300 km domain relative to the storm centre, from 0.5 km to 9 km above the
ground provided by TC-RADAR (Fig. 5). The PDF distribution clearly shows that all three simulated wind distributions skew
toward higher intensities compared to the observed data at both times. However, it is evident that 'AOW' reduces the
overestimation, particularly in the upper tail, indicating that 'AOW' improves the wind bias during the storm's development.
While the TC-RADAR provides comprehensive observations in both horizontal and vertical dimensions, the lowest level of
TC-RADAR for this storm is 0.5-km above the ground. This height limits us to validate modelled winds at heights that
hurricanes pose actual risks to offshore infrastructure and human activities. Dropsonde observations from aircraft can bridge
this gap. Figure 6 shows the vertical cross-sections of observed and simulated wind speed along the blue line shown in Fig.
3d. Consistent with TC-RADAR observations at higher altitudes, the dropsonde observation also shows that the strongest
winds are on the eastern side, approximately 10 to 30 km from the storm centre, with intensity gradually decreasing toward
the edge. On the other hand, much weaker speeds are observed on the western side, ranging from -10 to -200 km (Fig. 6a).
The observed patterns are reasonably captured in all three simulations, though they are generally overestimated. The
azimuthally averaged vertical profiles of simulated wind speeds in the inner-eyewall (defined as region within $0.2 \leq r/RMW \leq 1$)
and the outer-eyewall ($2 \leq r/RMW \leq 2.5$) regions are also evaluated (Fig 7). In both the inner- and outer-eyewall regions, it is
evident that all three simulations overestimate wind speeds in the low troposphere (below 2 km). However, 'AOW' aligns
more closely with observations compared to the other two simulations in both the inner- and outer-eyewall regions. Notably,
in the outer-eyewall region, 'AOW' is much closer to the observed values. These insights are particularly relevant for offshore
wind resources, as accurate wind profiles at hub heights and below are crucial for optimizing turbine placement and enhancing
energy generation efficiency in storm prone areas. A better representation of wind profiles, especially in the low troposphere
and near the surface, not only helps in predicting potential impacts on the turbines but also informs better design and operational
strategies.





**Figure 4. NOAA WP-3D airborne Doppler radar-observed (TC-RADAR) wind speeds at the 1-km level are shown in the first row, along with the corresponding model-simulated wind speeds for Hurricane Henri (2021) from the 'A' simulation (second row), 'AO' simulation (third row), and 'AOW' simulation (fourth row) at 00 UTC on August 21 (first and third columns) and 00 UTC on August 22 (second and fourth columns). The vertical cross-sections of wind speeds along the line from point A to B, indicated in the leftmost two columns, are presented in the rightmost two columns. All horizontal distributions are displayed in a 300 km × 300 km storm-centred coordinate.**



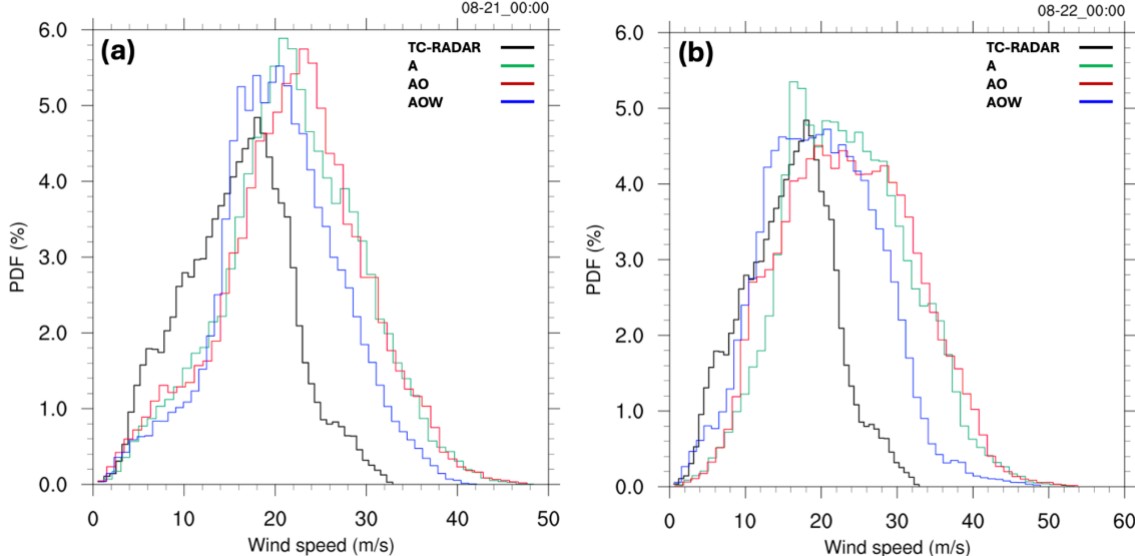

**Figure 5. Probability density function of wind speed in a 300 km x 300 km storm-centered coordinate, considering vertical levels**
**from 0.5 km to 9 km above the ground, for 00 UTC on 21 August (a) and 22 August (b) 2021. The data are derived from TC-RADAR**
**(black lines), experiment 'A' (green lines), experiment 'AO' (red lines), and experiment 'AOW' (blue lines).**




**Figure 6. Vertical cross-sections of (a) observed wind speed along the locations of the seven dropsondes, shown as colored dots in Fig. 3(d). The times of the first and last dropsondes are 23:23 UTC on 21 August and 00:11 UTC on 22 August, respectively. Corresponding cross-sections from (b) experiment 'A', (c) experiment 'AO', and (d) experiment 'AOW' are shown.**





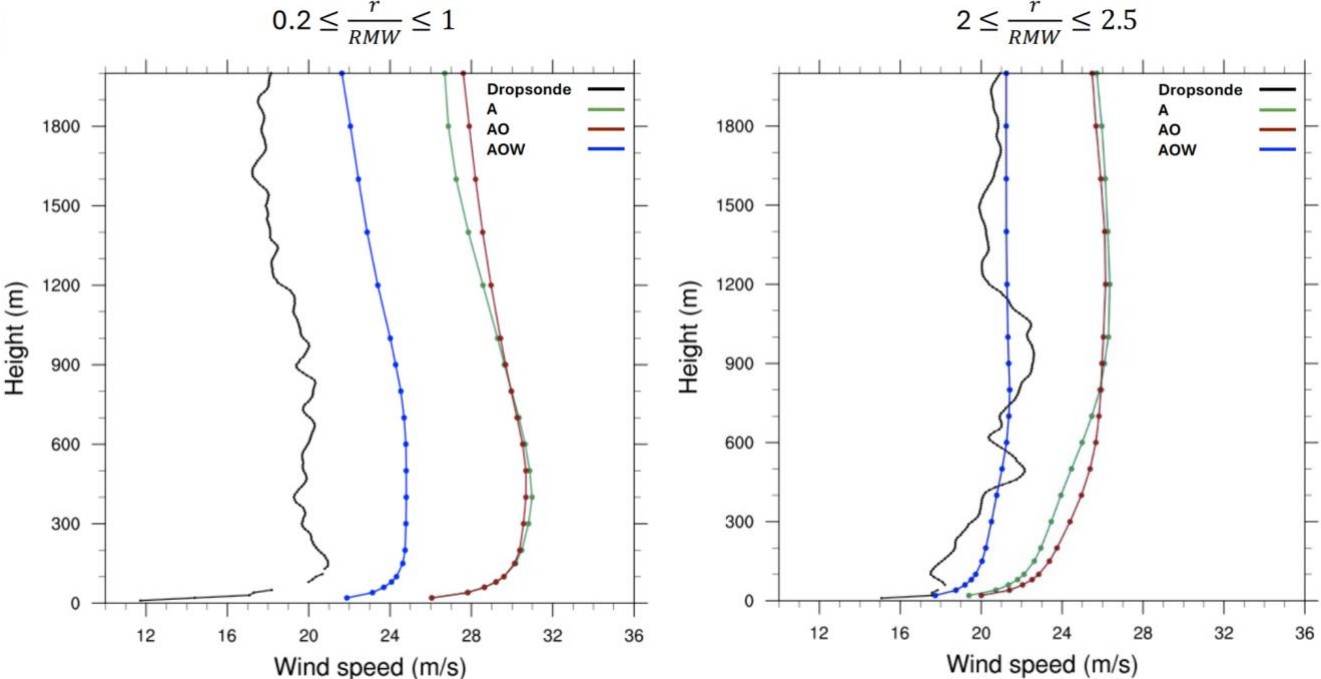


**Figure 7. Vertical profiles of azimuthally averaged wind speed for dropsondes (black lines), experiment 'A' (green lines), experiment 'AO' (red lines), and experiment 'AOW' (blue lines). The vertical profiles are azimuthally averaged in the inner-eyewall region (left; 0.2≤r/RMW≤1) and the outer-eyewall regions (right; 2≤r/RMW≤2.5) on 22nd August 2021.**

**4.2 Sea Surface Temperature and Waves**

**4.2.1 Sea Surface Temperature**

Several factors influence TC intensity, with SST and ocean surface roughness being among the most important, as they directly affect the heat and moisture available to fuel the storm (Zambon et al., 2014, 2021; Zhao et al., 2022). In our model configurations, the primary difference between the simulations lies in the treatment of SST and sea surface roughness, which ultimately impacts surface enthalpy and momentum fluxes through atmosphere-ocean interaction feedback. Therefore, we use SST as both a primary indicator and a driving mechanism of storm intensity to analyse the differences in Henri's intensity modelled by the three experiments.

Figure 8 illustrates the distribution of SST over the entire simulation domain for all three simulations, along with OSTIA observation at 12 UTC on 20th August (pre-storm) and 12 UTC on 23rd August (post-storm). The figure also shows the differences of SST between post- and pre-storm periods. Overall, all three simulations reasonably represent the SST distribution in both pre- and post-storm periods, effectively capturing the intensity and spatial extent of the Gulf Stream and surrounding warm SST zones (Fig. 8). Notably, both 'AO' and 'AOW' simulations adequately resolve cooler SSTs along the storm tracks, with comparable RMSEs and pattern correlations in comparison with SST in 'A' (Table 2). However, the two coupled simulations tend to overestimate cooling as the storms approach the U.S. Northeast Coast (Fig. 8f,i,l).



Figure 9 displays the distribution of SST for the four datasets in a 300 km x 300 km storm-centred at 24-hour intervals from
12 UTC on 20th August to 12 UTC on 22nd August. The area-averaged SST in the storm-centred coordinate is calculated and
OSTIA and listed in Table 3. At 12 UTC on 20th August, Henri turned northward and continued to strengthen with the aid of
relaxed wind shear and high SST associated with the Gulf Stream (Pasch, 2022; Fig. 9a). OSTIA well illustrates the warm
SST, with values exceeding 302.2 K (29.05 °C) on the southern side of the storm (Fig. 10i). However, none of the three
simulations accurately capture the southern warm SST distribution due to differences in their translation speeds and
corresponding locations, as well as biases in SST (Fig. 3a and 8-9). In addition, the two coupled simulations overpredict the
cold wakes on both southern and eastern sides of the storm (Figs. 9c-d), resulting in weaker minimum SLP compared to the
observation at this time (Fig. 3b). As the storm moved north-northeastward, the Gulf Stream and the surrounding warm waters
allowed it to deepen gradually, reaching its peak minimum SLP around 12 UTC 22nd August. It then weakened rapidly as it
interacted with land masses, making landfall in Rhode Island (Figs. 3 and 8). While the warm waters and cold wakes associated
with the storm are generally captured in all three simulations, the magnitude and spatial extent of the storm produced cold
wakes are mostly weaker, leading to an over-intensification of all simulated storms compared to the observation until they
make landfall (Figs. 9e-l). The overestimations are also partly due to the biases in track, making the simulated storms approach
the coast along different paths, which affect their intensity and interaction with the surrounding environment. Note that SST
in 'A' is updated by GFS reanalysis data at 6-hour intervals. Although the SST in GFS is derived from observation-based
reanalysis data, its distribution and magnitude differ from those of OSTIA (Fig. 8). This discrepancy likely arises from
variations in observational data sources and spatial resolution. OSTIA SST is generated using a combination of satellite
observations and in situ measurements (e.g., buoys, ships), offering high-resolution analysis at a 1/20° grid. In contrast, GFS
SST features a coarser resolution of 0.25° and predominantly relies on global ocean models and reanalysis datasets, which
may not incorporate the same observational data sources as OSTIA. Although all three simulations overestimate the intensity
of Henri, the experiment 'AOW' noticeably reduces the overestimation during the development as well as weakening stages,
as previously discussed. This might be due to the greater cooling of SST associated with wave-induced vertical mixing,
bringing cold water up to the surface consistent with prior studies (e.g., Wada et al., 2010; Zambon et al., 2014; Figs. 10d and
h).





**Figure 8. SST comparisons for various experiments: 'A' (SST updated by GFS at 6-hour intervals, first row), 'AO' (atmosphere-ocean model coupling, second row), 'AOW' (atmosphere-ocean-wave model coupling, third row), and OSTIA SST observation (fourth row). The first column shows SST at 12 UTC on 20th August (pre-storm), the second column displays SST at 12 UTC 23rd August (post-storm), the third column presents change in SST between pre-storm and post-storm. The black dots and lines indicate the best track derived from IBTrACS. The light blue dots and lines depict simulated storm locations and tracks.**




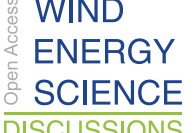

**Figure 9. Distribution of SST in a 300 km × 300 km storm-centered coordinate at 12 UTC on August 20th (top row), 12 UTC on August 21st (middle row), and 12 UTC on August 22nd (bottom row). The first column shows SST derived from OSTIA, the second column presents SST values from 'A', the third column displays SST from 'AO', and the fourth column shows SST from 'AOW' simulation.**

**TABLE 2. Temporally averaged root mean square error (RMSE) and Pearson product- moment coefficient of linear correlation (r) for SST in each simulation compared to OSTIA SST observations from 12 UTC on 20th August to 12 UTC 23rd August, 2021.**

| Experiment | RMSE | Pattern Correlation |
|---|---|---|
| A | 0.631 | 0.992 |
| AO | 0.564 | 0.991 |
| AOW | 0.577 | 0.990 |

**TABLE 3. Spatially averaged SST (K) derived from A, AO, AOW, and OSTIA observation in a 300 km x 300 km storm-centered coordinate at 12 UTC 20 August, 12 UTC 21 August, and 12 UTC 22 August.**

| Experiment | 08-20_12:00 | 08-21_12:00 | 08-22_12:00 |
|---|---|---|---|
| A | 302.01 | 301.81 | 299.37 |





| AO | 301.79 | 301.83 | 298.47 |
|---|---|---|---|
| AOW | 301.76 | 301.70 | 297.70 |
| OSTIA | 302.15 | 301.43 | 296.76 |

**4.2.2 Ocean Surface Waves**

In this section, we examine the accuracy of modelled ocean surface waves throughout the evolution of Hurricane Henri at the two NDBC buoys. During the storm's main development stage, station 41001 experienced the passage of the eye and the high winds and waves associated with the eyewall. Meanwhile, station 41002 was positioned about 120 km to the left of the storm's centre during its closest approach in the early development stage. The fully coupled experiment reasonably captures the general temporal trends of wind speed at both locations (Fig. 10b-c). However, due to the slower translation speed of the modelled storm, particularly from 06 UTC to 12 UTC on 21st August, the wind speed peaks are approximately 12 hours later than observed at the two locations. Consequently, the wave peak times at both locations are similarly delayed. The modelled magnitude of significant wave height is about 1-2.5 m higher than observed during peak times (Figs. 10d-e). At station 41001, the difference in wind speed between observed and modelled ones generally accounts for the difference in significant wave height. On the other hand, at station 41002, an additional factor that may contribute to the difference between observed and modelled significant wave height is the faster translation speed following the storm's slow movement from 06 UTC on 20th to 00 UTC on 21st August. Specifically, the translation speed of modelled storm is approximately 6.3 m s$^{-1}$, whereas the observed speed is approximately 4.8 m s$^{-1}$ during the period from 00 UTC to 06 UTC on 21st August. On the right side of the storm's path, the wind speed is amplified because the storm's forward motion adds to the wind speed. In this context, a faster moving storm can lead to stronger wind forcing, which increases wave energy and promotes greater wave growth, resulting in higher waves (Chen et al., 2013). While the fully coupled experiment effectively captures the wave direction at station 41001, it does not resolve the sharp directional change observed at station 41002 between 06 UTC and 09 UTC on 21st August. This lack of sharp directional change could be due to the model's increased wavelength and height as previously described, which prevent the observed rapid shifts in wave direction. Despite some biases in wave features, including magnitude and timing of peak wave height, the overall performance of the modelled waves at the two NDBC locations are reasonable and demonstrates the model's ability to capture general trends in storm-induced wave behaviour during Hurricane Henri.



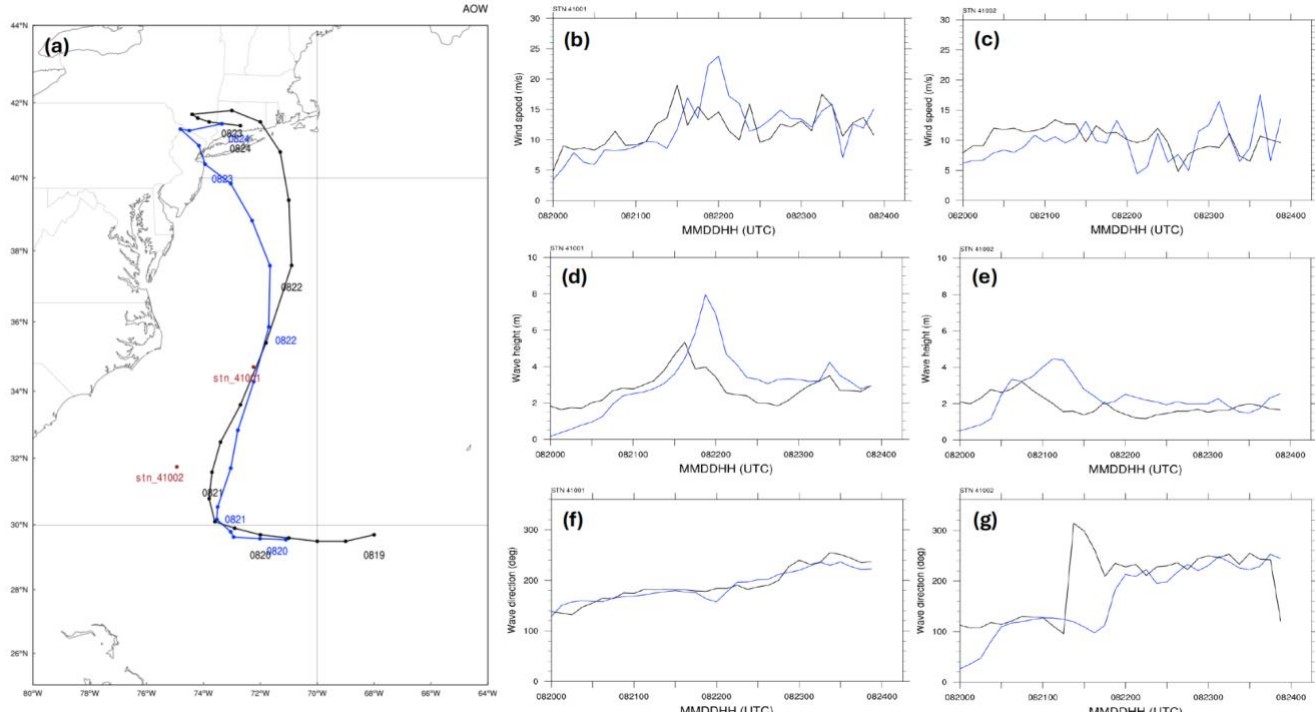

**Figure 10. Comparison of 'AOW' simulated in blue (a) track, (b)-(c) wind speed (m/s), (d)-(e) significant wave height (m), and (f)-(g) wave direction of Hurricane Henri with the observations in black from station 41001 (b),(d),(f) and station 41002 (c),(e),(g) during the period from 00 UTC on 20 August to 00 UTC on 24th August 2021.**

**4.3 Mechanisms Underlying the Improvement in the Fully Coupled Experiment**

So far we have learned that, compared to experiments 'A' and 'AO', 'AOW' not only reduces the overestimation of storm intensity (represented by the minimum SLP, Fig. 3) but also improves storm-scale wind structure (Fig. 4) and wind speed distribution (Fig. 5) compared to observations from the near surface to the upper troposphere for Hurricane Henri (2021). To examine the mechanisms behind these improvements, we first analyse SST and surface enthalpy flux of 'AOW' and compare them with those of 'AO' to examine ocean surface wave-induced processes and their influence on the evolution of Henri. We opted out of experiment 'A' in this analysis, as it is an atmosphere standalone simulation and does not consider atmosphere-ocean interactions.

Since both 'AO' and 'AOW' simulations have very similar storm tracks and comparable translation speeds, we are able to isolate the surface properties, including effects of SST, enthalpy flux, and surface roughness length on storm dynamics and evaluate how these factors contribute to differences in storm intensity and evolution. First, from a momentum transfer perspective, ocean surface waves characterize the surface roughness length ($Z_0$) of the ocean and regulate the exchange of momentum, in addition to heat and moisture, between the atmosphere and ocean. Without coupling the waves and the atmosphere, $Z_0$ or drag coefficient ($C_d$) is calculated solely based on wind speed (e.g., Charnock formulation). As a result,





'AO' might not accurately capture the dynamic interactions between the lower atmosphere and ocean surface during TCs
where wave effects and varying sea surface conditions significantly impact momentum transfer and overall system evolution.
We chose the time of 12 UTC on 22nd August 2021 to investigate these interactions in detail, focusing on how the inclusion of
wave dynamics in the fully coupled experiment 'AOW' alters the estimate of surface roughness length and the subsequent
effects on momentum and heat exchange during the hurricane's intensification phase. The timing is particularly significant, as
it marks the moment when the simulated storms from both 'AO' and 'AOW' reached their peak minimum SLP, occurring
about 12 hours prior to landfall. As shown in Fig. 11, 'AO' and 'AOW' simulate very different distributions of $Z_0$, clearly
demonstrating that 'AO' is solely a function of surface wind speed, while the $Z_0$ distribution of 'AOW' is distinct from the
surface wind distribution. This implies that wave dynamics play a crucial role in shaping $Z_0$ (Figs. 11a,b,e,f).
The impact of including ocean surface waves is not limited to just the representation of $Z_0$. 'AOW' is associated with stronger
winds, but lower SST, and surface enthalpy flux compared to 'AO.' The primary process responsible for cooling SST under
TCs is ocean vertical mixing, as discussed. During such events, the storm's surface winds create stress on the ocean surface
through friction, generating ocean currents in the mixed layer and momentum flux at the bottom of the atmosphere, leading to
evaporation from the ocean surface. In addition to the evaporative cooling, vertical velocity shear of the currents in the upper
ocean leads to turbulence, which mixes and entrains colder water from below the mixed layer, and reduces SST (e.g., Zhou et
al., 2023). This process is represented in both 'AO' and 'AOW.' However, the inclusion of wave dynamics in 'AOW' adds
additional vertical mixing through the following processes: the storm's surface winds build waves at the sea surface, and the
momentum transfer from the atmosphere to the ocean grows and propagates these waves. When the waves break, momentum
is transferred downward into the ocean currents, leading to vertical shear and thus vertical mixing. On the other hand, mixing
induced by non-breaking waves penetrates much deeper, leading to a further reduction in SST, as previously discussed. This
additional wave dynamics included in 'AOW' results in greater cooling in SST, leading to a reduced surface enthalpy flux
compared with 'AO' (Fig. 11). The reduced $Z_0$ in 'AOW' corresponds to a lower $C_d$, meaning that the roughness—primarily
due to the inclusion of waves—is less, which leads to higher wind speeds at the surface.
An important question remains regarding the discrepancy between minimum SLP and maximum wind speed in 'AO' and
'AOW.' Specifically, while 'AO' shows more intense (i.e., lower) minimum SLP than 'AOW', its maximum wind speed is
weaker compared to 'AOW' at this time (12 UTC on 22nd August; Fig. 3b-c and Figs. 11a-b) despite being linked to greater
surface enthalpy and momentum flux. As discussed, 'AO' is linked to higher $Z_0$ due to the absence of active wave dynamics
in the simulation, resulting in reduced surface wind speeds from increased frictional drag. This can lead to stronger subgradient
winds, where the actual wind speed is lower than what would be expected from the gradient wind balance. The relationship
between tangential circulation and radial inflow in the boundary layer is described by the agradient force (Montgomery and
Smith, 2014; Smith et al., 2009), which is defined as the difference between pressure gradient force and the sum of centrifugal
and Coriolis forces in the form of





$$Agradient\ Force\ (AF) = -\frac{1}{\rho}\frac{\partial p}{\partial r} + \frac{V_t^2}{r} + fV_t \qquad (3)$$

where $P$ is air pressure, $r$ represents radius from the TC centre, $V_t$ refers to tangential wind speed, and $\rho$ is air density. Near the surface, both 'AO' and 'AOW' deviate from gradient wind balance due to the effects of friction. Friction reduces the tangential wind speed, thereby weakening both Coriolis and centrifugal forces, while the pressure gradient force remains unchanged (negative agradient force: AF < 0). This imbalance results in a net inward force, driving inflow in the lower atmospheric layers (the secondary circulation). The magnitude of this inflow can be seen as an indicator of the deviation from the gradient wind balance, with stronger inflow corresponding to a greater degree of subgradient wind. The azimuthally averaged radial wind speed clearly shows that the enhanced inward flow towards the storm centre in the boundary layer in 'AO' compared with the 'AOW' (Fig. 12). In the boundary layer, stronger radial inflow transports additional absolute angular momentum (AAM) toward the storm's core, although friction disrupts the perfect conservation of AAM. As air moves inward, its radius decreases, causing tangential wind speeds to increase (as per the conservation of AAM). While friction within the boundary layer slows the tangential winds, the winds still strengthen near the core due to the influx of air masses. This increase in wind speed amplifies the outward centrifugal force, which is primarily counteracted by a greater inward pressure gradient force, resulting in a lower central pressure in the storm's core. In 'AO', even though the simulated storm is associated with stronger radial inflow through the process described above, the tangential winds along the storm centre are unable to increase relative to 'AOW' due to the additional friction effects over the ocean (Figs. 11 and 12). This extra friction in 'AO' is caused by an unrealistically calculated , which is only a function of surface wind speed. As surface wind speeds up, it generates more friction or drag on the ocean surface, which can further disrupt the conservation of AAM in 'AO.' As a result, the additional frictional dissipation of AAM in 'AO' leads to a reduction in the amount AAM available to drive wind's acceleration. Consequently, the wind speeds in 'AO' do not increase as much as expected despite the enhanced radial inflow. This explains the slower rate of intensification observed in 'AO' compared with its fully coupled counterpart, 'AOW' during the period from 06 UTC to 12 UTC on 22$^{nd}$ August. Note that as both simulated storms move through the baroclinic zone at this time, they may experience vertical wind shear. This can disrupt the typical inflow-outflow structure of tropical storms, leading to anomalous inflow at upper levels observed in Fig. 12.



528

**Figure 11. Distribution of (a)-(b) 10-m wind speed (m s-1), (c)-(d) surface enthalpy flux (W m-2), and (e)-(f) surface roughness length (m) derived from the experiment 'AO' (left column) and the experiment 'AOW' (right column) at 12 UTC on 22nd August 2021. All distributions are displayed in a 300 km × 300 km storm-centred coordinate.**





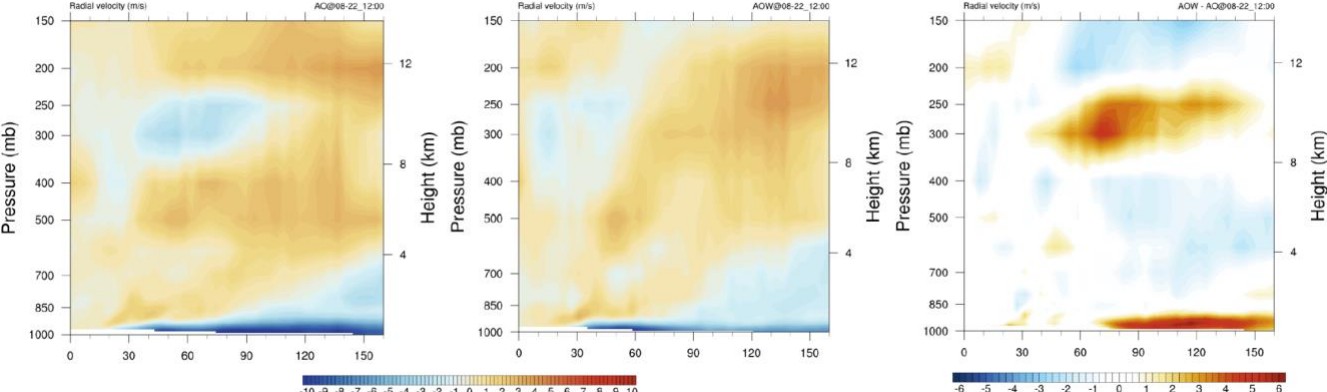

**Figure 12. The azimuthally averaged radial wind speed (m s$^{-1}$) for the (a) experiment 'AO', (b) experiment 'AOW', and (c) difference between 'AOW' and 'AO' at 12 UTC on 22$^{nd}$ August 2021.**

## 5 Implication for Potential Risks to Offshore Wind Energy

As the global demand for renewable energy continues to rise, offshore wind energy has emerged as a promising solution in the transition toward sustainable power generation. However, this opportunity comes with potential risks, particularly from TCs, which can generate extreme sea surface wave conditions, high wind speeds, and significant shear and veer between the ocean surface and hub heights (Wang et al., 2024a,b). Loads on offshore structures can arise from both aerodynamic and hydrodynamic forces and both act simultaneously on a turbine during a TC. In a design context, factors such as sustained wind speed, its relationship to wind gusts, the assumed vertical profile (shear) of the boundary layer, and wave heights and periods are crucial for calculating loads and are generally well understood. However, properties of the wind profile, including veer, as well as the temporal variability and directional dependence of wind and wave loads, remain less understood and are often not fully accounted for (Sanchez Gomez et al., 2023; Wang et al., 2024b).

### 5.1 Wind Veer

Wind veer is defined as the rate of change in wind direction with altitude (e.g., Churchfield and Sirnivas, 2018; Sanchez Gomez et al., 2023). Although the International Electrotechnical Commission (IEC) standards outlines the atmospheric conditions for weather extremes, including TCs, to guide the design of onshore (61400-1 IEC, 2019a) and offshore (61400-3 IEC, 2019b) wind turbines, wind veer is not accounted for in current design specifications (Sanchez Gomez et al., 2023). This omission remains despite its potential on turbine performance (Bardal et al., 2015; Gao et al., 2021) and loads (Churchfield and Sirnivas, 2018; Kapoor et al., 2020; Robertson et al., 2019; Sanchez Gomez et al., 2023). Large changes in wind direction with increasing altitude, driven by extreme events, can be destructive for wind turbines. For instance, a disruption in the grid connection caused by these extreme events may prevent the turbines from yawing into the wind, or the wind direction may change too quickly for the yaw control system to respond effectively, resulting in increased mechanical loads on the turbine components and





potential damage to the structure. To evaluate the wind veer simulated by 'A', 'AO', and 'AOW' simulations, we estimated
wind veer by calculating the difference in wind direction at multiple hub heights ranging from 100 m to 200 m in 10-m intervals
relative to the bottom (z = 30 m) of the turbine rotor layer. These results were then compared with dropsonde observations at
three different locations relative to the storm centre (~91 km to the left: yellow circle - called point A, ~40 km to the right:
green circle - called point B, and ~104 km to the right: red circle - called point C), as shown in Fig. 3d.
Firstly, the observations show that wind veer varies along the radius of the storm, with the veer noticeably increasing as it
approaches the centre of Henri around 00 UTC on 22 August. For example, at 150 m hub height, wind veer is 0.028°/m at
point A (to the left of the centre), 0.09°/m at point B (the closest point to the centre), and 0.06°/m at point C (to the right of the
centre). In addition, it is apparent that wind veer is greater on the right side of the storm compared to its left side counterpart
at all hub height levels (Figs. 13a-c). All three simulations significantly underestimate the wind veer, especially on the right
side of the storm (Figs. 13b-c). Furthermore, the greater veer observed on the right side of the storm in the dropsonde data is
not clearly captured in any of the simulations. Nevertheless, all three simulations do reasonably capture the general trend that
wind veer increases as it approaches the centre of the storm. Azimuthally averaged wind veer at 150 m relative to 30 m at 00
UTC on 22nd August, shown in Fig. 13d, clearly displays that the wind veers associated with the simulated storms gradually
decrease with radius in all three simulations. Among the three simulations, 'AOW' better matches the magnitude of the veer
at all levels at the three dropsonde locations although it still largely underestimates the values (Fig. 13). The largest veer across
the radius is seen in 'AOW,' consistent with the Figs. 13a-c.
Wind veer and shear can be influenced by several factors, with atmospheric stability, surface friction, and subsequent dynamic
and thermodynamic processes, such as both mechanical- and buoyancy-driven turbulence and vertical mixing (e.g., Englberger
and Lundquist, 2020; Murphy et al., 2020), all of which play key roles. For example, in stable atmospheric conditions, wind
veer and shear are typically more pronounced within the boundary layer, as stratification inhibits vertical mixing. In contrast,
under unstable conditions, enhanced turbulence promotes mixing, which can reduce the magnitude of wind veer and shear by
redistributing momentum. Additionally, surface friction slows near-surface winds, reducing their speed and altering their
direction, which creates a vertical gradient in wind speed and direction that contributes to wind veer and shear. In the light of
this, we examine these factors to understand how the three simulations differ in simulating these properties and how they affect
the representation of wind veer. Over the ocean, surface enthalpy flux represents the heat and moisture exchanged at the
atmosphere-ocean interface, with a lower flux suggesting a more stable lower boundary layer, while a higher flux is associated
with a less stable one. Regarding turbulence, turbulent kinetic energy (TKE) is commonly used as a proxy for turbulence in
the atmospheric boundary layer, representing the energy associated with turbulent motions. Thus, we use TKE to assess the
intensity of turbulence in the simulated storms. Figure 14 indicates horizontal distributions of surface roughness length, surface
enthalpy flux, and TKE at 30-m above the ground for all three simulations at 00 UTC on 22nd August. Consistent with our
previous findings, $Z_0$ in 'A' and 'AO' exhibit similar magnitudes and distributions, as both are driven by surface wind. In
contrast, $Z_0$ in 'AOW' is significantly weaker and displays a distinct spatial distribution, characterized by ocean wave processes
(Figs. 14a-c). Similarly, the surface enthalpy flux in 'AOW' is the weakest among the three, aligning with our previous findings





(Figs. 14d-f). In other words, 'AOW' is in a condition where decreased surface roughness and weaker surface enthalpy flux
act to suppress turbulent mixing. In relatively unstable atmospheric conditions, such as in 'A' and 'AO,' turbulence is more
pronounced due to buoyancy-driven mixing, which tends to redistribute momentum more evenly. Further, stronger $Z_0$ leads to
greater velocity shear between the atmosphere near the surface and above. This shear creates turbulent eddies that mix the
atmosphere. Therefore, more surface friction implies that the wind near the surface slows down more, creating stronger
turbulence that mixes the boundary layer. Figure 14 supports this idea, demonstrating that the experiments 'A' and 'AO' are
associated with greater surface enthalpy, stronger $Z_0$, and higher TKE compared to those in 'AOW.' Strong turbulence
generated by both mechanically and thermodynamically tends to reduce wind veer by mixing momentum, while weaker
turbulence likely allows the veer to persist within the boundary layer (e.g., Sanchez Gomez and Lundquist, 2020; Stull 1988).
Thus, the weaker turbulence linked to the lower $Z_0$ in 'AOW' may restrict vertical mixing near the hub heights and below,
allowing the wind veer to remain more pronounced and closer to the observed values compared to the other two simulations.
It is important to note that the 3 km grid spacing used in the atmospheric model is still too coarse to accurately resolve fine-
scale turbulence processes. For instance, Li et al. (2021) highlighted that mesoscale models are incapable of properly capturing
small-scale features such as roll vortices, which are large turbulent eddies commonly found in a hurricane's boundary layer.
Furthermore, Müller et al. (2024) discussed how the lower wind veer values simulated in mesoscale modeling during Typhoon
Megi, compared to those reported by Sanchez Gomez et al. (2023), could be attributed to the higher resolved wind veer
variability in large eddy simulations (LESs). This limitation likely contributes to misrepresenting wind veer magnitude
(underestimations) in all three simulations when compared to the observations. Nevertheless, this finding underscores the
ability to capture critical atmosphere-ocean interactions, such as cold wakes, momentum transfer, surface stress, and boundary
layer dynamics, particularly in the context of wind veer assessment. This suggests that relying solely on atmospheric-only
models to quantify wind veer, as previously studied, may lead to inaccuracies and underestimations, underscoring the
importance of incorporating atmosphere-ocean-wave interactions in future simulations.



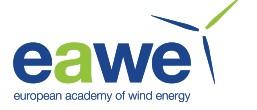


**Figure 13. Wind veer at multiple hub heights (ranging from 100 m to 200 m in 10-m intervals) relative to the bottom of the turbine rotor layer (z = 30 m) for (a) point A, (b) point B, and (c) point C. Point A is represented by a yellow dot, point B by a green dot, and point C by a red dot in Fig. 3d. (d) Azimuthally averaged wind veer (° m⁻¹) for the experiments 'A', 'AO', and 'AOW' at 00 UTC on 22nd August 2021. In (d), RMW denotes the radius of maximum wind, and r represents the radius relative to the storm centre.**

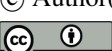

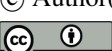

**Figure 14. Distribution of surface roughness length (m; upper panel), surface enthalpy flux (W m⁻²; middle panel), and turbulent kinetic energy (m² s⁻²; middle panel) for the experiments 'A' (left column), 'AO' (middle column), and 'AOW' (right column) are shown at 00 UTC on 22nd August 2021.**

## 5.2 Wind-Wave Misalignment

Wind-wave misalignment is another critical risk for offshore infrastructure that atmospheric-only models cannot estimate given

the lack of interaction between the atmosphere and ocean surface. Wind-wave misalignment can cause increased structural



loads on offshore wind turbines (discussed below), leading to fatigue damage and reduced operational lifespan. Figure 15
displays ocean surface wave information, including significant wave height, wave direction, and wavelength and 10-m wind
vector associated with the simulated in 'AOW' at 12 UTC on 22nd August. Previous studies have shown that the storm-induced
wave field around a hurricane is asymmetric, with the highest waves, as measured by significant wave height, typically
observed in the front-right quadrants of the storm (e.g., Chen et al., 2013; Wright et al., 2001). This typical characteristic is
also evident in our simulation, showing the highest significant wave heights in the right and front-right quadrants (Fig.15a).
Henri was heading northwest at 12 UTC on 22nd August 2021. In a moving storm, the waves to the right of the storm tend to
grow over time (Chen et al., 2013). This happens because the waves on the right side have a longer distance to travel and grow,
compared to the waves on the left side of the storm (Figs. 15a-b). In addition, directional misalignment of winds and waves is
evident on all sides of the storm, except the right side, consistent with prior study (Chen et al., 2013; Figs. 15d). This
misalignment is clearly represented in Fig. C1, which illustrates the time series comparison between surface wind direction
and mean wave direction at two NDBC buoy stations, 41002 and 41002. The figure highlights how the wave directions deviate
from the wind direction, as clearly observed in both NDBC buoys and the 'AOW' simulation, indicating complex interactions
at play.
These findings have significant implications for offshore wind energy operations and maintenance. When wind and wave
directions are not aligned (i.e., when they come from different directions), this creates substantial relative motion between
different parts of the wind turbine, specifically between the root (base) and the hub. This misalignment can lead to increased
movement or strain between these components. In contrast, when wind and waves are aligned, they combine in a way that
generates the highest impact velocities. This indicates that the forces acting on the turbine are stronger when the wind and
waves are moving in the same direction. In such conditions, with the two aligned, the turbine faces more severe impacts and
is at a higher risk of failure. Some studies have examined wind fields brought by TCs and their impacts on offshore wind
turbines (e.g., Sanchez Gomez et al., 2023; Wei et al., 2017; Itiki et al., 2023). However, most of these studies use either
atmospheric only models or empirical, parametric models that do not fully capture the complex interactions between wind and
wave forces during extreme events. This limitation makes it challenging to accurately predict the operational risks faced by
offshore turbines located in the hurricane belt. To assess this aspect, the models that incorporate atmosphere, ocean, and waves
components, are essential. Our fully coupled model with the three components can provide a realistic representation of wind
and wave behaviours, helping to predict wind-wave misalignment effectively. This allows for better assessment of the forces
acting on offshore wind turbines, enabling more informed design decisions and improved operational strategies to enhance the
longevity and reliability of wind turbine infrastructure.







**Figure 15. The fully coupled model simulated the following at 12 UTC on 22nd August 2021: (a) significant wave height (shaded; in meters) and wave propagation direction (vector), (b) mean wavelength (shaded; in meters) and 10-m wind (vector; in m/s), (c) 10-m wind speed (shaded; in m/s) and wind vector (in m/s), and (d) wave propagation direction vector (blue) and 10-m wind vector (red). The figures are displayed in a 300 km by 300 km storm-centered coordinate system. A reference wind vector of 20 m/s is shown in panels (b) and (c).**

## 6 Summary and Discussion

In this study, we developed a fully coupled modelling system (C-WFS) utilizing WRF, FVCOM, and SWAN to realistically

capture atmosphere-ocean-wave feedback on TC development and explore its implications for offshore infrastructure, such as





offshore wind turbines. We evaluated the performance of this coupled modeling system using Hurricane Henri (2021), selected
for its impact to the densely populated U.S. Northeast and nearby offshore wind lease area, as well as the extensive airborne
observations available. Three experiments with increasing complexity in atmosphere-ocean-wave coupled exchange
processes—'A', 'AO', and 'AOW'—were conducted and validated against a diverse range of observations, including IBTrACS,
airborne Doppler radar, dropsonde data, as well as both in-situ and satellite-based SST and wave measurements. The results
show that, while all simulated storms overestimate intensity in terms of minimum SLP, the fully coupled simulation 'AOW'
reduces this overestimation during both the development and weakening stages. Improvements are also evident in the 3-D
storm structure, where 'AOW' more accurately represents wind profiles across the entire atmosphere, including at low altitudes,
where the actual risks to offshore wind energy infrastructure occur. The enhanced performance of the fully coupled model is
primarily attributed to ocean wave-induced mixing, which leads to further cooling of the SST. Additionally, the reduced surface
roughness length and lower drag coefficient associated with atmosphere-ocean-wave interactions in 'AOW' simulation help
maintain a more realistic dynamical representation of the storm structure. In contrast, excessive friction and surface roughness
length driven by simplified parameterization (i.e., Charnock relation), in 'AO' simulation, result in increased frictional
dissipation of AAM as surface winds strengthen. This additional frictional dissipation, caused by the unrealistically driven
surface roughness length over the ocean, weakens tangential wind acceleration, thereby limiting the intensification of the storm
during its peak (from 06 UTC to 12 UTC on 22nd August 2021). As a result, 'AO' shows weaker storm wind speeds despite
having a more intense minimum SLP compared to 'AOW' during this period. This suggests that incorporating wave dynamics
in 'AOW' plays an important role in accurately simulating TC behaviour, ultimately enhancing predictive capabilities for storm
intensity and structure.
Additionally, the fully coupled 'AOW' experiment is characterized by weaker surface enthalpy, leading to a more stable
atmospheric boundary layer, reduced surface roughness, and lower TKE, all resulting from an improved representation of
dynamic and thermodynamic processes through coupled interactions. This leads to a more realistic simulation of wind veer,
with values closer to observations compared to the other two simulations. This finding indicates that, depending on location,
the coupling of ocean and waves can significantly affect wind veer, which is not considered in the current IEC standard
(Sanchez Gomez et al. 2023). Moreover, the model effectively captures wind-wave misalignment in comparison to buoy
observations. This misalignment poses a critical risk to offshore infrastructure, which atmospheric-only and atmosphere-ocean
coupled models are unable to predict.
Although we used Category 1 Hurricane Henri to validate the newly developed fully coupled model (C-WFS) and to highlight
the impact of coupling processes on the intensity, structure, and evolution of TCs, the same framework has also been tested
for higher-category hurricanes, including Laura (2020), a Category 4 storm that underwent rapid intensification. The results
reveal similar trends, with ocean waves contributing to a decrease in hurricane intensity. However, consistent with prior
research (e.g., Yamaguchi et al., 2017; Zhao et al., 2022), the atmospheric-only model tends to underestimate the intensity of



high-category TCs, particularly for those with minimum SLP near or below 940 hPa. As a result, the fully coupled model
further underestimates the intensity of these high-category storms due to additional wave-induced ocean mixing and
subsequent cold wakes. Some previous studies (e.g., Zhao et al., 2017, 2022; Zweers et al., 2015) suggested a promising
remedy for this issue, such as incorporating sea spray parameterization. For instance, Zhao et al. (2017, 2022) successfully
reproduced the intensities of Typhoons Megi (2010) and Haiyan (2013), both Category 5 equivalent super typhoons, using a
fully coupled model that incorporated sea spray parameterization. They demonstrated that sea spray increases the enthalpy
flux at the atmosphere-ocean interface, leading to a warmer boundary layer and a more unstable surface layer, which, in turn,
provide positive feedback for TC intensification. On the other hand, a recent study (Barr and Chen, 2024) examined the role
of sea spray in TC dynamics, showing that its effects are dependent on the storm's intensity. For weaker TCs, such as Category
1 storms, sea spray tends to inhibit intensification due to evaporative cooling in the boundary layer, acting as negative feedback.
However, as a TC strengthens (e.g., Category 2 or higher), increased spray production begins to contribute positively by
warming the boundary layer and enhancing deep convection near the eyewall. This transition highlights spray's dual role:
initially opposing intensification in weaker storms but eventually supporting rapid intensification in stronger ones, particularly
major hurricanes. While this study is highly informative and pioneering in demonstrating spray's dual role, its hypothesis is
based on only four TCs, limiting the generalizability of its conclusions. The precise impact of sea spray on TC structure and
intensity remains an open question, warranting further research across a broader range of TC events. In this study, sea spray
parameterization is not included in the C-WFS modelling system. However, as part of our ongoing research, we are integrating
sea spray into the system to investigate its impact on TC behaviour across various storm intensities. Advancing this work
represents a crucial step toward improving TC simulations and deepening our understanding of the associated enhancements.

Another aspect that remains unclear is the impact of the horizontal resolution of the ocean components on TC development
within the atmosphere-ocean coupled modelling framework. While the sensitivity of the atmospheric model's resolution to TC
representation is well-established, with general consensus suggesting that reducing horizontal grid spacing improves the
accuracy of storm intensity predictions (e.g., Gentry and Lakmann, 2010; Taraphdar et al., 2014; Prein et al., 2015), less is
understood about how the resolution of ocean components influences TC development. Higher ocean resolution allows for a
more detailed representation of mesoscale and submesoscale features (e.g., eddies and fronts affecting SST patterns, Zhang et
al., 2023) and their associated atmosphere-ocean interactions, such as heat fluxes, momentum transfer, and upper-ocean mixing
processes. These features likely play a critical role in modulating storm-induced SST cooling, redistributing ocean heat content,
and influencing the energy supply to TCs. As previously discussed, C-WFS employs an unstructured mesh grid, enabling
seamless transitions between coarse and fine resolutions. This approach removes the need for nested grids, commonly used in
existing fully coupled models (e.g., COAWST), which can introduce boundary artifacts. Consequently, C-WFS is uniquely
equipped to investigate how varying horizontal ocean resolutions affect coupling dynamics and storm development—an area
that will be thoroughly explored in future studies.





***Code and data availability.*** The WRF model (Version 4.5.1) is described by Skamarock et al. (2019), and its code is publicly
available from https://github.com/wrf-model/WRF (University Corporation for Atmospheric Research, 2019). The code for
FVCOM (Version 4.3.1., Chen et al., 2003, 2013) for ocean circulation model is publicly available at
https://github.com/FVCOM-GitHub/fvcom. The SWAN (Version 41.01, Booij et al., 1999) is a third-generation spectral wave
model developed at Delft University of Technology that computes random, short-crested wind-generated waves in coastal
regions and inland waters (http://swanmodel.sourceforge.net/). HYbrid Coordinate Ocean Model (HYCOM; Cummings and
Smedstad, 2014) analysis data used for ocean model forcing is available at http://hycom.org/dataserver/. NCEP provides
Global Forecast System (GFS; NCEP, 2015) data, which is used as atmospheric forcing data, available at
https://www.nco.ncep.noaa.gov/pmb/products/gfs/. The OSTIA (Good et al., 2020) global sea surface temperature provides
daily maps of foundation sea surface temperature at 0.05° × 0.05° available from
https://data.marine.copernicus.eu/product/SST_GLO_SST_L4_REP_OBSERVATIONS_010_011/description. The NCL and
Python codes for performing analysis and visualization are available at https://www.ncl.ucar.edu/ and
https://www.python.org/downloads/, respectively. All simulation data are available from the authors upon request.

**Appendix A: Radial profile of wind speed from the airborne Doppler radar and the three model simulations**

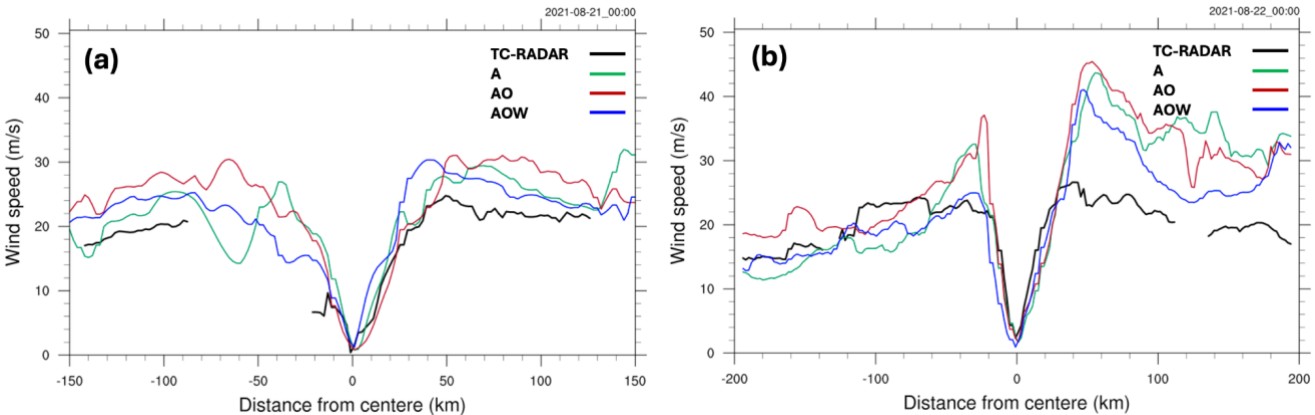


**Figure A1. Radial profiles of wind speed from the airborne Doppler radar and the three model simulations are shown in Figure 7**
**for (a) 00 UTC on 21st and (b) 22nd August 2021. The profiles are presented at the 1-km level along the line from A to B, as indicated**
**in Figure 4.**

**Appendix B: Averaged SST and surface enthalpy flux for 'AO' and 'AOW' in storm-centred coordinate**
Figure B1 displays the time series of spatially- averaged SST and surface enthalpy flux in a 300 km x 300 km storm-centred
coordinate. Both the time series reveal differences between the two coupled simulations, indicating that 'AOW' is associated
with cooler SSTs that are closer to the observation, as well as lower surface enthalpy flux over the entire simulation period.
This greater cooling of SSTs observed in 'AOW' partly explains the reduction in intensity. SST is reduced by the storm through
wave-induced vertical mixing and vertical mixed layer depth bringing cold water upward, which in turn reduces TC
intensification. Wang et al. (2024b) discussed wave-induced mixing primarily caused by wave breaking and non-breaking
wave orbital motion (non-breaking wave). Through their comprehensive literature review, they suggested that wave breaking-
induced mixing typically has a limited impact on SST and heat fluxes near the surface, and thus on TC intensity. In contrast,





non-breaking wave-induced mixing penetrates much deeper, enhancing vertical mixing and mixed-layer depth, which ultimately contributes to a greater reduction in TC intensification. Given this discussion, the inclusion of the non-breaking wave feature in 'AOW' may play an important role in moderating the thermal structure of the upper ocean, influencing the exchange of heat and moisture fluxes between the ocean and the atmosphere, and ultimately improving the intensity and structure of Henri.

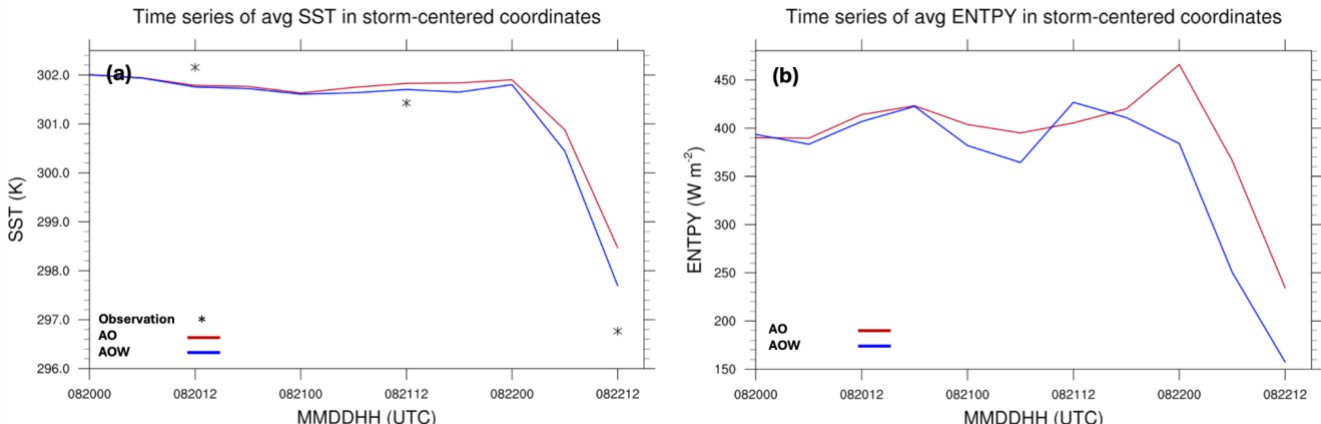

**Figure B1. Time series of spatial averaged (a) SST and (b) surface enthalpy flux in a 300 km × 300 km storm-centered coordinate. Data is derived from OSTIA observation (asterisks), experiment 'AO' (red lines), and experiments 'AOW' (blue lines).**

**Appendix C: Simulated and observed surface wind direction and mean ocean wave direction**

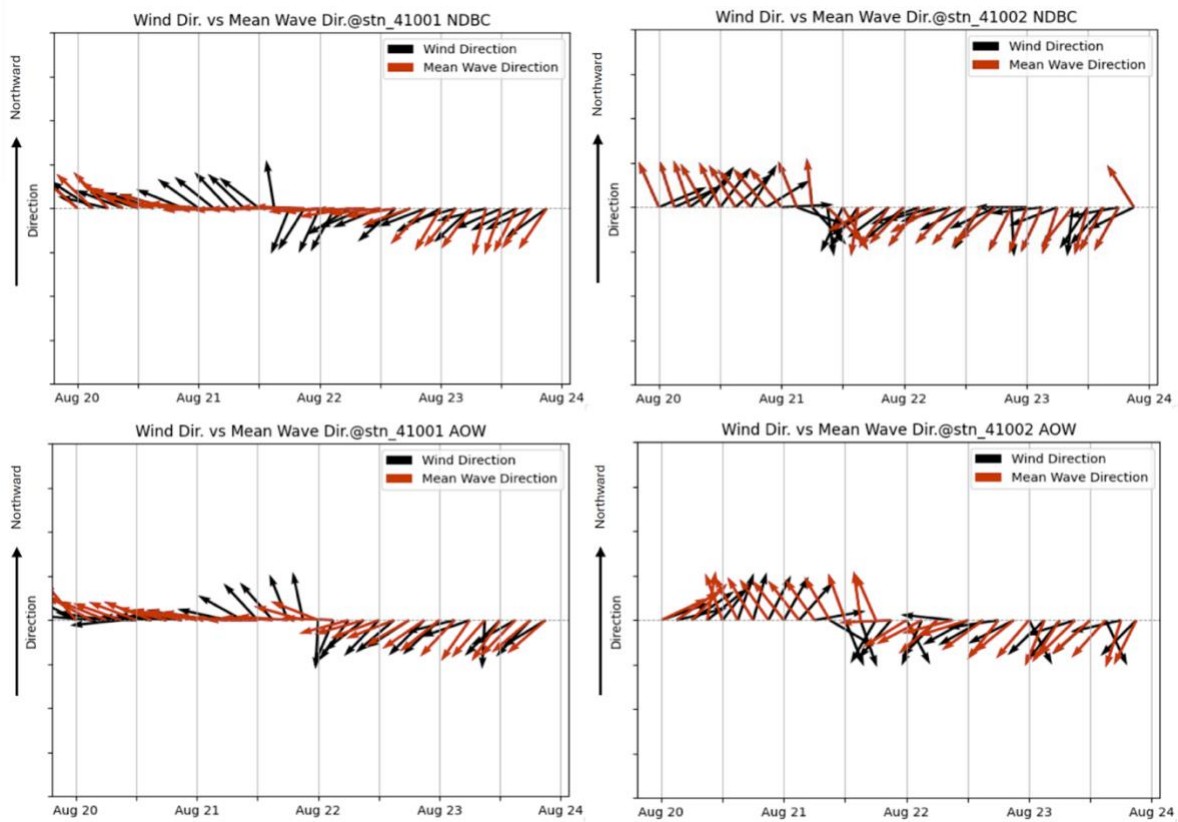

**Figure C1. Time series comparison of surface wind direction and mean ocean surface wave direction at two NDBC buoy stations, 41001 (left column) and 41002 (right column), derived from NDBC buoys (top panel) and experiment 'AOW' (bottom panel).**

***Author contribution.*** Conceptualization, Formal analysis, Validation, Visualization: CJ, JW, PX, CH, WP; Data curation, Investigation, Software: CJ, JW, CH, MB, GN; Funding acquisition, Resources, Supervision: JW, PX, WP; Methodology: CJ, CH, WP; Project administration: JW, PX; Writing – original draft: CJ, JW, PX, WP; Writing – review & editing: CJ, JW, PX, CH, MB, GN.

***Competing interests.*** The authors declare that they have no competing interests.

***Acknowledgements.*** This study is supported by the Wind Energy Technologies Office (WETO) of the U.S. Department of Energy (DOE) Office of Energy Efficiency and Renewable Energy. The WRF model was made available by the National Center for Atmospheric Research, which is sponsored by NSF. High-Performance Computing support from the Theta cluster operated by Argonne Leadership Computing Facility (ALCF) and Kestrel operated by National Renewable Energy Laboratory (NREL).

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
