# Peer review of "Fully Coupled High-Resolution Atmosphere-Ocean-Wave"

_Wind Energy Science, 2025_

## Referee Comment (RC1)

**Review: *Fully Coupled High-Resolution Atmosphere-Ocean-Wave Simulations of Hurricane Henri (2021): Implications for Offshore Load Assessments**

**General Comments**

This study develops a framework for two-way coupling atmosphere-ocean-waves called C-WFS. It provides improvements over existing frameworks such as COAWST including that the ocean model is on an unstructured grid so that no boundary artifacts from nesting appear at the ocean-atmosphere interface, and the inclusion of non-breaking waves. The study looks at a single hurricane case and compares atmosphere-only, atmosphere-ocean coupled, and atmosphere-ocean-wave coupled cases. The analysis shows little improvement in hurricane track as was expected by the authors (though it appears the C-WFS simulation produces the worst track), but reduces the high-bias of wind speeds over the ocean found in many atmosphere-only hurricane simulations. With that said, the C-WFS model still largely over-predicts wind speeds at all levels and regions of the storm for this case. The paper shows SST differences between a derived SST product (OSTIA) and the simulations with, again, some subtle improvements to the SST field in some instances, but large inaccuracies in others (Fig. 9a and 9d). The C-WFS model largely reduces the surface roughness due to the coupling of the wave model yet wind speeds are reduced in the C-WFS simulation. The reasons for this were not apparent to this reviewer. The only wind energy specific sections consider wind veer over a theoretical rotor swept area and wind-wave misalignment. All simulations showed very poor performance for wind veer, and there were no results for wind-wave alignment in the main part of the paper (the appendix has a figure for a single simulation).

This paper, like the review, is very very lengthy. It would benefit from significant editing or a re-write. Some of the findings seem to be exaggerated in their importance or significance based on what the plots present to the reader. The topic as a whole is relevant to wind energy but the paper lacks sufficient wind energy specific analysis and discussion. For these reasons and the reasons below, I am recommending to **reconsider the paper after major revisions.**

**Major Revisions**

This paper is very long and has several sections with excessive amounts of irrelevant information. Overall, the paper is not very well organized, the figures are illegible or inappropriate for the data being presented, and the findings seem to be exaggerated in instances based on the data being shown. There are no glaring "major" revisions, but the list of minor revisions is significant to the point that an overhaul of the paper text and figures is necessary.

Additionally, as the paper stands, there does not seem to be a sufficient link to wind energy as presented in the manuscript. Wind veer and wind-wave misalignment are the two sections that

address wind energy concerns and they are extremely weak. Without significant improvements to these sections, it is unclear why this paper would show up in Wind Energy Science.

I applaud the author's design of a new atmosphere-ocean-wave coupled framework and look forward to seeing the results of more cases, and a more rigorous analysis pertinent to wind energy.

**Minor Revisions**

- L60 - wave impacts on albedo immediately stuck out to me in thinking "how would albedo affect things below a hurricane in which the clouds would block most light from reaching the surface." Does this refer to the waves outside of the hurricane, or within it?
- L92 - my understanding is that convection-permitting resolutions for the atmosphere are typically 4-km or less, so it is strange to see the use of 5-km grid spacing and tout the importance of convection-permitting resolution. This is also mentioned directly in this paper on L198.
  See also:
  https://doi.org/10.1175/1520-0493(1997)125<0527:TRDOEM>2.0.CO;2
  https://doi.org/10.1002/2014RG000475
- L97-109 - COAWST is mentioned and the model framework is stated to be different, but I'm having trouble distinguishing which of the components are different from COAWST. For example, the regional refinement is clear, but then does COAWST not include non-breaking waves (the sentence begins with "in addition" so I'm led to believe COAWST does not include this either)? In line 105 the framework is said to be more realistic than statistical-parametric models - does COAWST use these for its parameterizations or are we no longer talking about COAWST here? I'd recommend restructuring the paragraph to make clear how the framework is different from COAWST specifically, and then maybe a new paragraph for the other benefits (if needed).
- L208 - 12 km seems much coarser than the ocean and atmospheric resolutions to simulate something that is very small in scale; waves. Could you provide evidence that this is a typical model grid spacing for wave simulations?
- L220-223 - are these supplementary results or just not shown? Please specify.
- L243-244 - makes sense. Do other studies also adjust dropsonde positions like this?
- L244-246 - "shown as blue and colour dots" reads strange. Consider listing the colors you want the reader to focus on (black is, afterall, a color), or changing the shapes of the ones you want the reader to focus on so you can state "shown as squares" or something like that.
- General model setup: GFS is used and it is mentioned that other IC/BCs were tested using reanalysis datasets, so was this the GFS final analysis data that was used here, or was this a reforecast of some sort? Additional clarification would be nice for the boundary conditions of each of the models to answer if this was a reforecast/hindcast or a simulation of the storm. Also, could this model be run operationally or do we lack the necessary boundary conditions for the ocean and waves? I now see on L400 that GFS "reanalysis" data is used. I do not think GFS offers a reanalysis product but instead

considers their product a "final analysis" product. It's a subtle, but significant distinction. GEFS has a reanalysis, so if that is what is used, it should be corrected.

- L336-349 - I cannot follow this comparison due to the coarseness of the observations in Fig. 6a. The observations and simulations seem entirely different. Consider a different plot for this comparison (vertical profiles seem like an obvious choice).
- L378-384 - this figure seems to show that A matches observations much better than AO and AOW with the ocean model adding a lot of warming to the surrounding environment. The paragraph seems to argue the AO and AOW models are similar to A and simply overestimate the cooling, but the figure shows differently. It is unclear how/where the statistics from Table 2 are calculated but they do not match the eye-test.
- L407 - Can this not be determined from the model as has been done in other studies? This framework is the novel aspect of the study, so showing that the C-WFS methodology is improving on answers due to specific aspects being resolved that other models miss should be included.
- L427-447 - comparisons with these buoys, particularly 41002, depict inaccuracies in the wave model. Buoy 41002 is only 10 grid cells away if I have it right that grid spacing is 12 km. Is it possible that the issues are due to too coarse of resolution in the wave model?
- L456-462 - this paragraph is leading in that it highlights the positive aspects of the "AOW" experiments but so far the improvements appear to be minor and in some instances it looks like AOW isn't really better at all (or in the case of storm track, is worse). While it makes sense to not include A since there are no ocean-atmosphere interactions, in some cases A appears to be the best or comparable simulation. This paragraph could be reframed to simply state that surface enthalpy flux will be examined for the atmosphere-ocean coupled models.
- L475-477 - Figures 11a and 11e have notable differences so saying that this clearly shows z0 from the AO simulation is solely a function of surface wind speed is inaccurate.
- L477 - "This implies…" isn't this something that other papers have already stated?
- L490-491 - there are studies that have suggested that surface roughness goes down in high winds such as hurricanes. It might be good to include some references to show that a weakening roughness with higher wind speeds has merit and is not nonsensical given that traditional knowledge of over-water roughness says that roughness increases with wind speed.
- L478-491 - the AOW simulation produces lower roughness *and* lower wind speeds. The last sentence in this section states that lower z0 leads to higher wind speeds. So, how can the roughness decrease be attributed to the decreased winds of the AOW simulation? Also, if you calculated Charnock's formulation of z0 for these wind speeds, you likely would see lower roughness values since the wind speeds are now reduced. There must be other effects here that are resulting in the reduced wind speed.
- L509-511 - this sentence (and following sentences) is about the boundary layer but the plots are showing up to 150 mb which makes the comparisons with the boundary layer (the lowest sliver of the plot) difficult to follow. The "anomalous" inflow at upper levels described in L526 appears to be the only reason for including such heights in the plot. I'm not sure it's worth it to sacrifice the clarity of a figure for a paragraph with several

lines dedicated to the boundary layer just for one short sentence about inflow at upper levels that isn't elaborated on or really proven to even be "anomalous."

- L492-526 - this paper is very long as-is, and this section in particular seems to take what could be a couple sentences and turns it into a page of text. Consider modifying for brevity.
- L546 - I'm not sure "wind veer" requires citations of a couple of papers that use it. It is well known.
- L552-555 - are these scenarios (particularly the first one) realistic? What would cause a hurricane to disrupt the connection of an offshore wind turbine? Do hurricanes typically see rapid wind direction changes? Additionally, this section is discussing veer - but both examples are of disconnectivity and wind direction change (which the IEC standards do cover). This section is only introductory, but it is poorly formulated.
- L555-557 - wind veer is the difference in wind direction with height, so this sentence is saying you are estimating wind veer by calculating wind veer.
- L556 - the model setup has 12 levels below 100 m, so is there interpolation being performed to get to exactly 10 m intervals in this layer, or is it roughly 10 m intervals?
- L560-571 - the results here are remarkably poor. Is it possible that the data are not taken from similar locations in the model? Particularly nearest to the eye of the storm, the performance seems unreasonably bad. Is it truly that well-mixed within the simulations?
- L604 - so Sanchez Gomez et al. 2023 was able to simulate reasonable levels of wind veer in a hurricane? How were they able to do this? Did they use a coupled ocean-atmosphere-wave model?
- L606-610 - this comes out of nowhere. The results that were just shown were very poor, but the last few sentences of this paragraph seem to claim that this framework does a good job. It also suggests that all prior studies relied solely on atmosphere-only simulations, though there have been numerous studies that couple atmosphere-ocean and even atmosphere-ocean-wave as has been cited within this paper already. In all, it seems that there is a study that did better at simulating wind veer in a storm (Sanchez Gomez et al 2023), but this isn't elaborated on or used to explain why C-WFS (and the individual members of the system) performs so poorly.
- L639-645 - this information requires citation.
- L642 - misalignment causes more strain; L644 wind alignment causes the turbine to face more severe impacts and is at a higher risk of failure. How do these conflicting ideas exist? There are no citations for such claims.
- L646-650 - the results so far do not indicate that including the impacts of oceans and waves are really "essential" and in some cases the atmosphere-only models appear to do just fine. Furthermore, they do have wave information in the models, it simply isn't coupled. Wind-wave alignment can be assessed in an atmosphere-only model if wave direction is provided as a boundary condition from something like ERA5 or GFS.
- Section 5.2 - this section provides no new insight or results. Wind-wave misalignment is simply discussed but there is no actual evidence supporting the usefulness of the C-WFS framework. Consider adding such analysis (best option) or removing the section entirely (last resort since it is argued that wind-wave alignment is important).
- L684 - a "more stable atmospheric boundary layer" was never actually shown. This was deduced from a lower surface enthalpy flux and lower TKE at a single level, but

atmospheric stability is a function of height and typically shown through atmospheric profiles or the calculation of quantities such as the Richardson Number.

- L686-687 - while the values are slightly closer to observations, they are overall very far off which suggests that there is something inherently missing or wrong in the modeling system as a whole. Is it worth highlighting marginal increases in performance in something that is overall simulated very poorly?
- L687-689 - these sentences are inaccurate and/or worded incorrectly. It reads as if saying ocean coupling isn't included in IEC standards. Additionally, it seems to make the claim that this is a novel finding of the paper.
- L689-690 - are the comparisons to buoy data shown in the paper? How do we know that it's more accurate than the other two models when only the AOW experiment was shown?
- L690-691 - after reading *this* paper, I'm not convinced of the critical risk of wind-wave misalignment. Additionally, both atmosphere-only and atmosphere-ocean models absolutely can simulate wind-wave misalignment through the boundary conditions.
- L711-712 - if the 4 cases in the sea spray study are insufficient for generalizable results, then this study with a single case also must not be presented as generalizable. There is no "limitations" section in this paper although there are certainly many to be mentioned; this is one of them.

**Technical Suggestions**

- L124 - not a big deal, but the C-WFS acronym might be helped by underlining the letters for which it represents (coupled WRF-FVCOM-SWAN).
- L180-182 - similarly here: the "A" experiment wasn't immediately apparent and I was thinking the next experiment would be "B" but it was "AO" and then I had to think a bit before registering "Atmosphere-Ocean." So it might be helpful to spell this out (e.g., "The WRF standalone simulation – representing the atmosphere only – is named experiment 'A'..."). It also may be worth considering aligning the experiment names with the acronym for the model framework (e.g., "WRF", "C-WF", "C-WFS") so that the "AOW" experiment is clearly the full C-WFS framework.
- L326 - why is this figure mentioned in the paper but included in the appendix?
- Figure 6 - the observations are difficult to compare against the model. Would profiles not be better? The x-axis is also poorly formatted.
- L386-387 - "The area-averaged… Table 3" - this sentence has something wrong grammatically. Possibly at "calculated and OSTIA and"
- Fig 10a never referenced.
- L519 - missing word. Z0?
- General comment: the section titles without punctuation are strange. For example, section 5 reads "5 Implication for Potential Risks…" as if there will be a list of 5 things.
- L560-563 - this refers to Fig 13, right?
- L625 - why is this discussed below? Consider just discussing this here to give context to the reader.

---

## Author Comment (AC1)

**Review: *Fully Coupled High-Resolution Atmosphere-Ocean-Wave Simulations of Hurricane Henri (2021): Implications for Offshore Load Assessments**

**General Comments**

This study develops a framework for two-way coupling atmosphere-ocean-waves called C-WFS. It provides improvements over existing frameworks such as COAWST including that the ocean model is on an unstructured grid so that no boundary artifacts from nesting appear at the ocean-atmosphere interface, and the inclusion of non-breaking waves. The study looks at a single hurricane case and compares atmosphere-only, atmosphere-ocean coupled, and atmosphere-ocean-wave coupled cases. The analysis shows little improvement in hurricane track as was expected by the authors (though it appears the C-WFS simulation produces the worst track),but reduces the high-bias of wind speeds over the ocean found in many atmosphere-only hurricane simulations. With that said, the C-WFS model still largely over-predicts wind speeds at all levels and regions of the storm for this case. The paper shows SST differences between a derived SST product (OSTIA) and the simulations with, again, some subtle improvements to the SST field in some instances, but large inaccuracies in others (Fig. 9a and 9d). The C-WFS model largely reduces the surface roughness due to the coupling of the wave model yet wind speeds are reduced in the C-WFS simulation. The reasons for this were not apparent to this reviewer. The only wind energy specific sections consider wind veer over a theoretical rotor swept area and wind-wave misalignment. All simulations showed very poor performance for wind veer, and there were no results for wind-wave alignment in the main part of the paper (the appendix has a figure for a single simulation).

This paper, like the review, is very very lengthy. It would benefit from significant editing or a re-write. Some of the findings seem to be exaggerated in their importance or significance based on what the plots present to the reader. The topic as a whole is relevant to wind energy but the paper lacks sufficient wind energy specific analysis and discussion. For these reasons and the reasons below, I am recommending to **reconsider the paper after major revisions.**

**Major Revisions**
This paper is very long and has several sections with excessive amounts of irrelevant information. Overall, the paper is not very well organized, the figures are illegible or inappropriate for the data being presented, and the findings seem to be exaggerated in instances based on the data being shown. There are no glaring "major" revisions, but the list of minor revisions is significant to the point that an overhaul of the paper text and figures is necessary.
Additionally, as the paper stands, there does not seem to be a sufficient link to wind energy as presented in the manuscript. Wind veer and wind-wave misalignment are the two sections that address wind energy concerns and they are extremely weak. Without significant improvements to these sections, it is unclear why this paper would show up in Wind Energy Science. I applaud the author's design of a new atmosphere-ocean-wave coupled framework and look forward to seeing the results of more cases, and a more rigorous analysis pertinent to wind energy.

We truly appreciate the reviewer taking the time to review and make constructive comments and questions, which have helped to strengthen and improve the manuscript.

First of all, we have significantly reduced the word count while preserving the core ideas and analyses. We have moved some information into a supplementary document. The main text now is reduced to ~7,500 words from ~11,500, excluding the title, abstract, and references.

Secondly, as the reviewer noted, the C-WFS model shows relatively poor performance in capturing the storm's track and intensity, which, in turn, results in biases in SST. As discussed in the original manuscript, additional experiments confirmed that applying spectral nudging significantly improves track and intensity simulations. However, we deliberately chose not to apply nudging in this study, because our primary goal is to isolate the specific effects of multi-factor coupling on Hurricane Henri's behavior. While nudging can improve forecast accuracy, it imposes external constraints that may mask or confound the intrinsic influence of air-sea interactions on storm dynamics. Rather than aiming for perfect agreement with observations, we prioritized understanding how varying degrees of coupling affect storm evolution, particularly wind field characteristics relevant to offshore wind turbines. Despite track and intensity biases across all three models, each reasonably captures the storm's overall structure, landfall timing, and general evolution. Such consistency allows us to meaningfully assess the distinct influence of atmosphere-ocean-wave coupling on storm behavior, which remains the core scientific question of our study.

Lastly, we acknowledge that our wind energy-specific analysis in this study is limited, as our primary objective is to introduce and demonstrate a new atmosphere–ocean–wave coupled modeling system, with a focus on how the inclusion of wave dynamics influences TC evolution, particularly wind structure. To date, atmosphere–ocean–wave interactions have been largely overlooked in the context of offshore wind energy risk assessment. This study aims to begin addressing that gap by providing initial insights into how realistic coupling affects TC wind structure, with potential implications for offshore wind system vulnerability. However, we agree that a comprehensive analysis is necessary to fully evaluate how this coupled modeling system impacts wind energy–relevant factors. We are currently working on a separate manuscript using this fully coupled model to conduct a more in-depth wind energy-specific analysis. This includes examining the coupling effects on factors both considered and not considered in the IEC standard, such as wind veer, shear, and turbulence, with the goal of informing improvements to the IEC standard for hurricane-prone regions and enhancing offshore wind turbine resilience.

This manuscript presents an initial framework and proof-of-concept, emphasizing the value of incorporating wave dynamics into atmosphere-ocean modeling. This work lays the foundation for more detailed investigations into offshore wind infrastructure risk and resilience. To maintain a focused scope and avoid overextending the current manuscript, we have removed the original implications section from the revised manuscript, with the intention of presenting a comprehensive wind energy-focused analysis in a dedicated follow-up study.

We hope that this clarification addresses the concern and demonstrates how this work serves as an important step forward improving offshore wind energy risk assessments by incorporating realistic atmosphere-ocean-wave interactions within the modeling framework. A detailed line-by-line response to the reviewer's comments is provided below.

**Minor Revisions**

1) L60 - wave impacts on albedo immediately stuck out to me in thinking "how would albedo affect things below a hurricane in which the clouds would block most light from reaching the surface." Does this refer to the waves outside of the hurricane, or within it?

The discussion by Cavaleri et al. (2012) addresses the effects of waves from a broader perspective beyond TC conditions. While they highlight that increased surface roughness generally lowers albedo by enhancing forward scattering of sunlight, this mechanism is unlikely to have a significant effect under TCs, where deep cloud cover already limits incoming solar radiation. Therefore, the role of wave-induced albedo changes in TC environments is expected to be negligible.

While this discussion offers useful context, the level of detail exceeds what's appropriate for the statement in line 60. Accordingly, we chose not to incorporate additional content into the revised main manuscript.

2) L92 - my understanding is that convection-permitting resolutions for the atmosphere are typically 4-km or less, so it is strange to see the use of 5-km grid spacing and tout the importance of convection-permitting resolution. This is also mentioned directly in this paper on L198. See also:
https://doi.org/10.1175/1520-0493(1997)125<0527:TRDOEM>2.0.CO;2
https://doi.org/10.1002/2014RG000475

We agree that a 5 km grid spacing falls outside the conventional convection-permitting range and lies within the so-called "gray zone," where some convective processes are partially resolved but still require cautious interpretation. In response, we have removed the original sentence in the Introduction to improve clarity and conciseness.

3) L97-109 - COAWST is mentioned and the model framework is stated to be different, but I'm having trouble distinguishing which of the components are different from COAWST. For example, the regional refinement is clear, but then does COAWST not include non-breaking waves (the sentence begins with "in addition" so I'm led to believe COAWST does not include this either)? In line 105 the framework is said to be more realistic than statistical-parametric models - does COAWST use these for its parameterizations or are we no longer talking about COAWST here? I'd recommend restructuring the paragraph to make clear how the framework is different from COAWST specifically, and then maybe a new paragraph for the other benefits (if needed).

Thank you for the suggestion. We have revised the paragraph to clarify which components distinguish our framework from standard COAWST and to separate this discussion from our comparison to statistical–parametric models. Please see lines 57-64.

To address your specific question: the original COAWST framework (as described in Warner et al., 2010) does not include non-breaking wave orbital motion (such as Stokes drift or vortex force effects) in its default configuration. It primarily incorporates breaking wave processes through radiation stress from SWAN and standard surface flux formulations. Xu et al. (2023) demonstrate a custom extension to a COAWST-based modeling system in which non-breaking wave-induced turbulence is explicitly added, highlighting that such processes are not present in the default COAWST code and require additional implementation.

Regarding the comment in line 105, the statement comparing to statistical–parametric models refers to a broader category of simpler, empirical models that do not resolve physical processes. COAWST, like our C-WFS framework, is a physics-based model and does not use statistical–parametric approaches. To avoid confusion, we have entirely restructured the paragraph in the revised Introduction.

4) L208 - 12 km seems much coarser than the ocean and atmospheric resolutions to simulate something that is very small in scale; waves. Could you provide evidence that this is a typical model grid spacing for wave simulations?

Waves are modeled in phase-averaged form (spectral wave models) such that the actual small physical scale of wind waves is unimportant (e.g., see Pringle and Kotamarthi, 2021). For spectral wave models 12 km is typically regarded as sufficient in deeper waters (even for hurricanes) and higher resolution (e.g, down to 1 km) is only needed for accurate modeling of the nearshore (e.g., Abdolali et al., 2020; Oladejo, et al., 2025). We modified the sentence to cite these previous studies demonstrating the reasonable choice of resolution for the open ocean.

5)  L220-223 - are these supplementary results or just not shown? Please specify

These simulations were conducted as part of sensitivity testing to assess the robustness of our results across different physical parameterizations and forcing datasets. As they serve a supporting role, the outcomes have been excluded from the manuscript and supplementary materials for brevity. This has been clarified in lines 152-154.

6)  L243-244 - makes sense. Do other studies also adjust dropsonde positions like this?

Yes, adjusting dropsonde positions relative to the model storm center is a commonly used approach in TC modeling evaluation studies (e.g., Creasey and Elsberry, 2017). This method allows for a more consistent comparison between observation and model data, especially when there are differences in storm position or track between the model and reality. We modified this sentence to cite the previous study in line 164-166.

7)  L244-246 - "shown as blue and colour dots" reads strange. Consider listing the colors you want the reader to focus on (black is, afterall, a color), or changing the shapes of the ones you want the reader to focus on so you can state "shown as squares" or something like that.

In Figure 3, the dropsonde locations used for the wind veer analysis were previously shown as colored dots. As discussed earlier, we decided to remove the implications sections and the wind veer analysis from the manuscript to focus on the core objective of the study and have a separate study for a comprehensive wind-energy related analysis. Accordingly, the statement in lines 244-246 has been removed, and the figure (Figure 3 in the revised manuscript) has been revised to reflect this change.

8)  General model setup: GFS is used and it is mentioned that other IC/BCs were tested using reanalysis datasets, so was this the GFS final analysis data that was used here, or was this a reforecast of some sort? Additional clarification would be nice for the boundary conditions of each of the models to answer if this was a reforecast/hindcast or a simulation of the storm. Also, could this model be run operationally or do we lack the necessary boundary conditions for the ocean and waves? I now see on L400 that GFS "reanalysis" data is used. I do not think GFS offers a reanalysis product but instead considers their product a "final analysis" product. It's a subtle, but significant distinction. GEFS has a reanalysis, so if that is what is used, it should be corrected.

Thank you for pointing this out. To clarify, we used the GFS analysis data provided at 0.25 by 0.25 grid spacing (https://rda.ucar.edu/datasets/d084001/) as atmospheric forcing for the simulations. This is an operational analysis product, not a reforecast or reanalysis dataset. We have corrected "reanalysis" to "analysis" throughout the revised text for accuracy and consistency. As discussed in Comment 5, other atmospheric forcing datasets (e.g., ERA5) were employed in internal sensitivity tests to evaluate the robustness of the model results. While these tests are not included in the manuscript, they helped reinforce our confidence in the findings presented. For oceanic initial and boundary conditions, we used the 1/12° HYCOM (Hybrid Coordinate Ocean Model) analysis data, as described in Section 3.1 (Cummings and Smedstad, 2014) The wave model is initialized from a quiescent state (added to revised manuscript in lines (and the model is initialized from a quiescent state in line 149-150.

9) L336-349 - I cannot follow this comparison due to the coarseness of the observations in Fig. 6a. The observations and simulations seem entirely different. Consider a different plot for this comparison (vertical profiles seem like an obvious choice).

The limited number of dropsonde observations (only seven) results in a coarse spatial distribution, which affects the interpretability of Figure 6a (now Figure S1). As noted, we acknowledge that this comparison may not be ideal in the main text.

To address this, we have moved the figure to the supplementary document, where it now serves as supporting information for Figure 7 (now Figure 6 in the revised manuscript). The azimuthally averaged vertical wind profiles presented in Figure 7 (now Figure 6 in the revised manuscript) offer a clearer and more robust comparison, highlighting the superior performance of 'AOW' simulation in representing the low-troposphere wind profile relative to others.

10) L378-384 - this figure seems to show that A matches observations much better than AO and AOW with the ocean model adding a lot of warming to the surrounding environment. The paragraph seems to argue the AO and AOW models are similar to A and simply overestimate the cooling, but the figure shows differently. It is unclear how/where the statistics from Table 2 are calculated but they do not match the eye-test.

To clarify, Table R1 (Table 2 in the revised manuscript) presents the temporally averaged RMSE and Pearson correlation for SST in each simulation, evaluated against the OSTIA SST dataset over four days from 12 UTC on 20 August to 12 UTC on 23 August 2020. Since OSTIA provides daily SST data, statistics in Table R1 (Table 2) represents an average over these four days (i.e., 20, 21, 22, and 23 August 2020). In Figure 8 (now Figure S3 in the revised manuscript), we presented SST distributions only at 12 UTC on 20 August (pre-storm) and 12 UTC on 23 August 2020 (post-storm) to highlight the spatial changes in SST associated with the storm translation. As such, this figure may not capture all the variability reflected in the temporally averaged statistics shown in Table R1 (now Table 2). To provide a more complete view, we include Figure R1 in the response, which shows the SST distribution from OSTIA (top panel), simulation 'A' (middle panel), and simulation 'AOW' (bottom panel) for all four days (20–23 August). We exclude simulation 'AO' here, as its SST distribution largely resembles that of 'AOW'.

Since GFS SST is technically generated by observed SST data, 'A' captures key large-scale features well, such as the Gulf Stream and warm waters along the Gulf Coast. However, it consistently underestimates SST across the domain, particularly in the open ocean regions of the North Atlantic during this period. In addition, its relatively low resolution limits its ability to capture small-scale SST patterns (Fig. R1 top and middle panels), which contributes to the higher RMSE values shown in Table R1 (Table S2).

The 'AOW', which is driven by oceanic initial and boundary conditions from the HYCOM analysis, successfully capture major SST features such as the Gulf Stream and Gulf Coast, with enhanced spatial detail. This improved representation contributes to lower RMSE values compared to the atmosphere-only simulation ('A') (Table R1). However, 'AOW' tends to overestimate SSTs in the open North Atlantic and underestimate them near the northeastern U.S. coast (Figs. R1 fourth column), likely due to cold wakes generated by the simulated storms and deviations in their tracks from observations. Nevertheless, the 'AOW' reasonably reproduce the observed SST, with RMSE values of 0.577—lower than that of 'A'— while maintaining comparable pattern correlation overall (Table R1).

We have revised the corresponding discussion in the manuscript to include these additional details. Furthermore, we have replaced Fig. 8 (now Fig. S3) with a new version that includes the 'AO' simulation,

as shown in Figure R1, to clarify the comparison. Please see the lines 267-280. The original figure has been moved to the supplement for reference.

[Figure]

**Figure R1. SST distribution (K) for OSTIA (upper panel), 'A' (middle panel), and 'AOW' (bottom panel) at 12 UTC on 20 (first column), 21 (second column), 22 (third column), and 23 August 2020 (fourth column). The black dots and lines indicate the best track derived from IBTrACS. The light blue dots and lines depict simulated storm locations and tracks.**

**Table R1. Temporally averaged root mean square error (RMSE) and Pearson product- moment coefficient of linear correlation (r) for SST in each simulation compared to OSTIA SST observations from 12 UTC on 20 to 12 UTC 23 August, 2021.**

| Experiment | RMSE | Pattern Correlation |
|---|---|---|
| A | 0.631 | 0.992 |
| AO | 0.564 | 0.991 |
| AOW | 0.577 | 0.990 |

11) L407 - Can this not be determined from the model as has been done in other studies? This framework is the novel aspect of the study, so showing that the C-WFS methodology is improving on answers due to specific aspects being resolved that other models miss should be included.

We thank the reviewer for this suggestion. We agree that investigating mechanisms such as wave-induced vertical mixing and their contributions to SST cooling is an important direction. The C-WFS framework is capable of capturing and quantifying these processes through the formulations outlined in Section 2, and indeed, these processes are well-established within the ocean dynamics and modeling community (e.g., Wada et al., 2010; Zambon et al., 2014). Furthermore, Figures R1 (Fig. 7), R2 (Fig.S5 in the revised

manuscript), and the accompanying discussion in the supplementary material clearly illustrate that incorporating wave effects leads to additional SST cooling in association with the storm.

While we do not claim that these processes are uniquely resolved in our modeling system, the C-WFS framework enables their effects to be incorporated and evaluated within a fully coupled atmosphere-ocean-wave modeling context. Importantly, the influence of such coupled processes on TC evolution, particularly with regard to boundary layer wind structure and distribution, and their implications for offshore wind energy infrastructure, has not been explicitly addressed in previous wind energy risk assessment under TC conditions.

This study aims to assess the influence of wave coupling within a fully coupled modeling system during a realistic, storm-evolving scenario. To maintain focus and accommodate space limitations, we have emphasized the role of wave coupling in modulating TC intensity and wind structure at the air–sea interface and within the planetary boundary layer, which are directly relevant to offshore wind applications.

[Figure]

**Figure R2. Time series of spatial averaged (a) SST and (b) surface enthalpy flux in a 300 km ✕ 300 km storm-centered coordinate. Data is derived from OSTIA observation (asterisks), experiment 'AO' (red lines), and experiments 'AOW' (blue lines).**

12) L427-447 - comparisons with these buoys, particularly 41002, depict inaccuracies in the wave model. Buoy 41002 is only 10 grid cells away if I have it right that grid spacing is 12 km. Is it possible that the issues are due to too coarse of resolution in the wave model?

We believe that wave model grid spacing has little impact on wave accuracy. Previous studies (e.g., Tolman et al., 2005) have shown that wave accuracy is highly sensitive to the quality of atmospheric forcing, particularly during hurricanes. This sensitivity helps explain the discrepancies observed in our wave model results shown in Fig. 10 (Fig. 8 in the revised manuscript). We attribute these differences primarily to wind forcing and the relative position of each buoy to the TC center. For instance, the overprediction of wind speed at buoy 41001 around 00 UTC on 22 August leads to an overestimation of wave height. Additionally, because the simulated TC track passes farther east than the observed track when approaching buoy 41002, the discrepancy in the timing of the wave direction shift can be attributed to this track error.

13) L456-462 - this paragraph is leading in that it highlights the positive aspects of the "AOW" experiments but so far the improvements appear to be minor and in some instances it looks like AOW isn't really better at all (or in the case of storm track, is worse). While it makes sense to not include A

since there are no ocean-atmosphere interactions, in some cases A appears to be the best or comparable simulation. This paragraph could be reframed to simply state that surface enthalpy flux will be examined for the atmosphere-ocean coupled models.

Thank you for the comment and suggestion. We acknowledge the reviewer's point that the improvements in 'AOW', while evident in storm intensity and structure, are not uniformly superior across all metrics. In particular, as noted, the storm track performance of 'AOW' (RMSE of 126.1 km) is slightly worse compared to 'A' (RMSE of 123.7 km). However, our intent in this section is to explore the physical mechanisms that may contribute to the reduced overestimation of storm intensity, especially wind in 'AOW.'

Using a combination of multiple observational datasets (IBTrACS, TC-RADAR, and dropsondes), we have demonstrated that 'AOW' shows notable improvements in capturing the three dimensional wind structure and intensity evolution compared to the other two simulations. In contrast, the 'A' simulation does not exhibit comparable skill in reproducing the three dimensional storm structure or intensity distribution (as shown in Figs. 4-6 and S1–S2 in the revised manuscript). In response to this comment, we have revised the paragraph to be more concise and focused on the key areas of improvement, namely, intensity and structural characteristics, while avoiding overstating the overall performance of 'AOW'.

14) L475-477 - Figures 11a and 11e have notable differences so saying that this clearly shows z0 from the AO simulation is solely a function of surface wind speed is inaccurate.

Thank you for pointing this out. To clarify, in 'AO' simulation, the model computes the surface roughness length ($Z_0$) using the Charnock relation (Charnock 1995; Chen et al., 2013):

$$Z_0 = \alpha \frac{\tau}{\rho_a g}$$

$$\tau = \rho_a C_d U_{10}^2$$

where, $\tau$ is wind stress, $\alpha$ is Charnock parameter (=0.0185), $g$ depicts gravitational acceleration, $\rho_a$ is air density, and $C_d$ is drag coefficient. Therefore, $Z_0$ in this configuration is ultimately dependent on surface wind speed. We have revised the sentence in lines 364-368 to clarify this point.

15) L477 - "This implies…" isn't this something that other papers have already stated?

Yes, previous studies (e.g., Taylor and Yelland, 2001; Drennan et al., 2005; Shimura et al., 2017) have indeed discussed the influence of wave dynamics on surface roughness, many of which build upon or parallel the framework proposed by Taylor and Yelland (2001). This sentence has been rephrased in lines 370-373.

16) L490-491 - there are studies that have suggested that surface roughness goes down in high winds such as hurricanes. It might be good to include some references to show that a weakening roughness with higher wind speeds has merit and is not nonsensical given that traditional knowledge of over-water roughness says that roughness increases with wind speed.

As the reviewer suggested, there is mounting observational and theoretical evidence that under extreme winds (e.g., in hurricanes), the relation between wind speed and surface roughness can change significantly.

Several studies have shown that the drag coefficient ($C_d$) saturates or even decreases once wind speeds exceed approximately 30-35 m/s, largely because of wave processes which dampen momentum transfer to the ocean (e.g., Donelan et al., 2004; Powell et al., 2003). Since surface roughness length ($Z_0$) is directly timed to $C_d$ via Monin-Obukhov theory (e.g., $C_d = (\frac{\kappa}{ln\,(\frac{10}{z_0})})^2$), this saturation implies a corresponding weakening or plateauing of surface roughness. We have added these references to support the physical plausibility of decreasing $Z_0$ in high winds in lines 373-375.

17) L478-491 - the AOW simulation produces lower roughness and lower wind speeds. The last sentence in this section states that lower z0 leads to higher wind speeds. So, how can the roughness decrease be attributed to the decreased winds of the AOW simulation? Also, if you calculated Charnock's formulation of z0 for these wind speeds, you likely would see lower roughness values since the wind speeds are now reduced. There must be other effects here that are resulting in the reduced wind speed.

To clarify, 'AOW' is associated with lower surface roughness and stronger wind speeds, while 'AO' exhibits the opposite, as discussed in the original manuscript (L478–491). We have thoroughly investigated the observed discrepancy between minimum SLP and maximum wind speed in the 'AO' and 'AOW' simulations. This analysis is framed in terms of the absolute angular momentum (AAM) budget and the influence of wave-induced surface roughness and is presented in detail in lines 391-431 of the revised manuscript. Please see the lines for a comprehensive explanation of the underlying dynamics.

Regarding the second comment, Z₀ in 'AOW' was calculated using the formulation of Taylor and Yelland (2001), shown in Equation (1). Unlike the other two simulations ('A', 'AO'), where Z₀ depends solely on surface wind speed through the Charnock relation, this formulation (Taylor and Yelland, 2001) incorporates wave parameters, making Z₀ responsive to the evolving sea state. This clarification has been added to lines 364-368 in the revised manuscript.

18) L509-511 - this sentence (and following sentences) is about the boundary layer but the plots are showing up to 150 mb which makes the comparisons with the boundary layer (the lowest sliver of the plot) difficult to follow. The "anomalous" inflow at upper levels described in L526 appears to be the only reason for including such heights in the plot. I'm not sure it's worth it to sacrifice the clarity of a figure for a paragraph with several lines dedicated to the boundary layer just for one short sentence about inflow at upper levels that isn't elaborated on or really proven to even be "anomalous."

We appreciate this comment and agree with the concern raised. In the revised manuscript, to improve clarity, we have removed the Figure 12 (in the original manuscript) and introduced a new figure (Figure 12 in the revised version) that focuses on the lower atmosphere and displays data up to 500 m, allowing for clearer visualization of boundary layer processes.

19) L492-526 - this paper is very long as-is, and this section in particular seems to take what could be a couple sentences and turns it into a page of text. Consider modifying for brevity.

Thank you for the comment and suggestion. As mentioned earlier, we have significantly reduced the manuscript length while maintaining the core ideas and highlighting the key results.

One important point we aim to demonstrate is that realistic modeling of wind structure requires not only atmosphere–ocean ('AO') coupling, but also fully interactive atmosphere–ocean–wave ('AOW') coupling. The discussion in lines 391–431 is part of our effort to highlight this need. Specifically, our analysis demonstrates that excluding wave influence ('AO') affects more than just surface heat flux exchanges; it also leads to excessive surface roughness due to the use of a simplified, wind-only parameterization (Charnock relation). This, in turn, causes greater frictional dissipation and results in

weaker tangential winds, despite a deeper central pressure compared to 'AOW'. These findings underscore the importance of incorporating wave dynamics for realistic and accurate forecasts of TC wind structure, with implications for offshore wind risk assessment. The revised discussion in lines 391-431 now provides a more comprehensive and focused explanation, with a clearer connection to the key physical mechanisms underlying the differences between the simulations between wave-inclusive ('AOW') and non-wave-inclusive ('AO') simulations.

20)  L546 - I'm not sure "wind veer" requires citations of a couple of papers that use it. It is well known.

This has been removed now. A comprehensive, wind energy-focused analysis is ongoing and will be presented in a dedicated follow-up study.

21) L552-555 - are these scenarios (particularly the first one) realistic? What would cause a hurricane to disrupt the connection of an offshore wind turbine? Do hurricanes typically see rapid wind direction changes? Additionally, this section is discussing veer - but both examples are of disconnectivity and wind direction change (which the IEC standards do cover). This section is only introductory, but it is poorly formulated.

Yes, these are realistic scenarios.

Regarding the grid disconnection, several studies have shown that offshore wind turbines rely on external or backup power sources to operate yaw motors and maintain proper alignment with incoming wind during extreme events. Kim and Manuel (2014) demonstrated that in hurricane conditions, loss of electric grid power or backup system failure, due to insufficient capacity, mechanical failure, or onshore grid disruption caused storm-induced extremes, can cause yaw control systems to become unresponsive, leading to potentially significant yaw misalignment and increased structural loading. Similarly, Rose et al. (2012) found that turbines lacking functional yaw control during hurricanes face a much higher risk of structural failure due to misalignment-induced loads.

Regarding rapid wind direction changes, observations and high-resolution simulations confirm that rapid wind direction shifts can occur in the hurricane eyewall and near-surface inflow. For instance, Kapoor et al. (2020) analyzed large-eddy simulations of a Category 5 hurricane and report instantaneous wind-direction changes of about 30° over 30 sec within the eyewall. GPS dropwindsonde profiles (Franklin et al., 2003) documented transient directional fluctuations of 10° – 20° on time scales of tens of seconds near the surface in the eyewall boundary layer. Worsnop et al. (2017) show that eyewall wind fields can exhibit directional shifts up to 50° in similarly short intervals. All these findings underscore the importance of capturing such rapid veer events in turbine design standards.

While we acknowledge that yaw misalignment and directional shifts are addressed in IEC standards, these cover the bulk orientation of the entire rotor relative to the dominant wind flow (mean horizontal wind direction) and scenarios where the turbine cannot adjust its orientation. On the other hand, Sanchez Gomez et al. (2023) highlighted that while IEC standards are robust in accounting for various extreme events and gross yaw misalignments, they may not explicitly or sufficiently detail design load cases that specifically address the complex, asymmetric loading induced by significant wind veer profiles under hurricanes. This is an active area of research where high-resolution simulations (at dozens of meters) are demonstrating effects that current design paradigms might simplify or overlook. In other words, this study points out a finer-scale omission regarding the vertical shear of wind direction (veer) and its precise load implications.

Given the length and the focus of our current study, we do not include this specific discussion in the revised version, but a more detailed and turbine-focused analysis is underway and will be presented in a dedicated follow-up study.

22) L555-557 - wind veer is the difference in wind direction with height, so this sentence is saying you are estimating wind veer by calculating wind veer.

We thank the reviewer for pointing this out. The phrasing could have been more precise. Our intention was to detail the specific methodology to quantify wind veer in this study. As noted previously, this discussion has now been removed from the revised manuscript.

23) L556 - the model setup has 12 levels below 100 m, so is there interpolation being performed to get to exactly 10 m intervals in this layer, or is it roughly 10 m intervals?

The model employs a terrain-following vertical coordinate system (eta levels), with 12 vertical levels below 100 m altitude. Due to the nature of this coordinate system, these levels are not evenly spaced in terms of geometric height. Therefore, to analyze variables at fixed heights (e.g., meters above ground level), interpolation from the native eta levels to constant-height levels is required. With that said, the fine vertical resolution within the lowest 100 m allows us to better capture find-scale atmospheric processes and, importantly, enables more accurate interpolation to constant-height levels within the layer.

24) L560-571 - the results here are remarkably poor. Is it possible that the data are not taken from similar locations in the model? Particularly nearest to the eye of the storm, the performance seems unreasonably bad. Is it truly that well-mixed within the simulations?

We agree that the simulations exhibit a significant underestimation of wind veer compared to the dropsonde observations. As discussed in our response to Comment 6, we carefully adjusted the dropsonde positions relative to the simulated storm center to ensure spatial consistency between the observed and modeled locations.

Given that all three simulations show consistent underestimation of wind veer, we believe this issue stems from inherent limitations associated with the model's horizontal resolution (3 km) and the use of a planetary boundary layer (PBL) parameterization. At this resolution, the model cannot explicitly resolve turbulent eddies that are critical for accurately representing the detailed vertical structure of the wind, particularly shear and veer, within the TC boundary layer (e.g., Chen et al., 2021).

PBL schemes parameterize the net effect of unresolved turbulent processes. However, these parameterizations often lead to excessive vertical mixing of momentum, which tends to smooth out strong vertical gradient and, consequently, underestimates wind veer (Brown et al. 2005; Zhang et al., 2017). This effect is particularly pronounced in extreme conditions like hurricane boundary layers where complex interactions between strong wind and stability dictate localized veer (Zhang et al., 2017). While 'AOW' improves surface interactions and processes, the fundamental treatment of unresolved turbulence at this resolution still leads to an over-mixed boundary layer.

Dropsonde data, being high-resolution point observations, capture local variability that is smoothed in the model's parametrization. Therefore, accurately resolving its precise magnitude and azimuthal asymmetry, as seen in observations, would likely require explicitly resolving turbulence through large-eddy simulation (LES) approaches.

The relevant discussion has been removed in the revised manuscript, and a more detailed and turbine-focused analysis is underway and will be presented in a dedicated follow-up study.

25) L604 - so Sanchez Gomez et al. 2023 was able to simulate reasonable levels of wind veer in a hurricane? How were they able to do this? Did they use a coupled ocean-atmosphere-wave model?

Sanchez Gomez et al. (2023) performed idealized large-eddy simulations of TCs using an atmosphere-only model, with a horizontal resolution of 55.55 m in the innermost domain. These idealized simulations could not be evaluated for accuracy due to their nature; rather, they were used to understand the behavior of simulated hurricane winds and how their features, such as veer, shear, and turbulence, can potentially exceed current IEC standards.

26) L606-610 - this comes out of nowhere. The results that were just shown were very poor, but the last few sentences of this paragraph seem to claim that this framework does a good job. It also suggests that all prior studies relied solely on atmosphere-only simulations, though there have been numerous studies that couple atmosphere-ocean and even atmosphere-ocean-wave as has been cited within this paper already. In all, it seems that there is a study that did better at simulating wind veer in a storm (Sanchez Gomez et al 2023), but this isn't elaborated on or used to explain why C-WFS (and the individual members of the system) performs so poorly.

We respectfully clarify that the intention of lines 606-610 is not to claim that 'AOW' perfectly captures wind veer or that it outperforms all existing modeling systems in an absolute sense. Rather, our point is to emphasize that, relative to 'A' and 'AO' where they lack wave induced dynamics and interactions with atmosphere, 'AOW' exhibits improvements in capturing key air-sea interaction processes that influence wind veer (e.g., stress feedback).

Furthermore, while we acknowledge the existence of prior coupled modeling studies cited in this manuscript, very few (if any) have directly focused on wind veer characteristics in the TC environment with an explicit offshore wind energy implication. To the authors knowledge, most existing studies instead focus on broader atmospheric processes or general air–sea interactions.

While Sanchez Gomez et al. (2023) presents a valuable analysis of yaw misalignment, turbulence, and wind veer during hurricanes, their study is based on an idealized setup and does not employ a fully coupled atmosphere–ocean–wave modeling framework, nor does it include a comparative analysis with uncoupled simulations.

This discussion has been omitted, and a comprehensive investigation will be presented in a dedicated follow-up study.

27) L639-645 - this information requires citation.

Previous studies (e.g., Shanahan and Fitzgerald, 2025; Verma et al., 2020) demonstrate that misalignment between wind and wave directions significantly affects the loading and structural behavior of offshore wind turbines. For example, Shanahan and Fitzgerald (2025) highlight the critical role of wind–wave misalignment in increasing structural risk for floating offshore wind turbines, showing that such misalignment significantly amplifies side-to-side turbine deflections and may lead to underestimation of fatigue loads by up to 50%. On the other hand, Ding et al. (2025) found that aligned wind–wave directions result in more severe fatigue over time, reducing structural reliability notably after 20 years of operation. These findings suggest that long-term exposure to aligned conditions requires more proactive structural monitoring and maintenance. Additionally, Verma et al. (2020) demonstrated that aligned wind–wave conditions can lead to more severe impact loads and failure modes during offshore turbine installation.

In the revised manuscript, the wind-wave misalignment analysis has been moved to Section 4.4, with corresponding discussion presented in lines 324-334.

28) L642 - misalignment causes more strain; L644 wind alignment causes the turbine to face more severe impacts and is at a higher risk of failure. How do these conflicting ideas exist? There are no citations for such claims.

We have added appropriate citations (e.g., Ding et al., 2025; Shanahan and Fitzgerald, 2025; Verma et al., 2020) to clarify the distinct risks associated with both wind-wave misalignment and alignment (as also noted in our response to Comment 27). These are not contradictory findings but rather reflect different types of structural vulnerabilities. Misalignment tends to amplify lateral deflections and dynamic loads, particularly in floating systems, while prolonged aligned conditions have been shown to lead to greater fatigue accumulation and reduced structural reliability over time. We have revised the text in the manuscript accordingly to clearly distinguish between these two mechanisms and their implications (lines 324-334).

29) L646-650 - the results so far do not indicate that including the impacts of oceans and waves are really "essential" and in some cases the atmosphere-only models appear to do just fine. Furthermore, they do have wave information in the models, it simply isn't coupled. Wind-wave alignment can be assessed in an atmosphere-only model if wave direction is provided as a boundary condition from something like ERA5 or GFS.

As discussed in our responses to Comments 13–17, we have demonstrated that the 'AOW' simulation improves the representation of storm wind structure, particularly with respect to boundary layer processes and near-surface wind fields. While the 'A' and 'AO' simulations may appear adequate in terms of track, MSLP, and maximum wind speed, these metrics alone do not reflect the full physical realism needed for applications such as offshore wind energy. As detailed in Section 5, the absence of wave coupling in the 'AO' simulation (this can be applied to the 'A' simulation as well) leads to inconsistent representations of surface roughness and air-sea interaction process (e.g., surface heat and moisture fluxes), which in turn affect the accuracy of low-level wind fields. Such limitations can result in misleading assessments of wind conditions that are critical for turbine design and structural safety.

Regarding wind-wave misalignment and alignment: while it is technically possible to prescribe wave direction in the 'A' simulation using boundary conditions from forcing datasets, this approach has critical limitations. Such prescribed wave fields are based on observed or reanalysis storm conditions. If the simulated storm differs somewhat in track, intensity, or timing, which is common, then the external wave data will no longer match the storm's actual evolution, creating inconsistencies between wind and wave fields. These discrepancies are especially problematic during rapidly evolving events such as TC. Moreover, as discussed in Section 5, these inconsistencies can impact storm development itself by modifying air-sea fluxes and surface roughness, ultimately leading to inaccurate storm associated wind fields. In contrast, a fully coupled modeling framework enables dynamic, two-way interactions between the atmosphere, ocean, and waves, ensuring wind and wave fields evolve coherently in space and time with the simulated storm. This consistency is "*essential*" for high-confidence assessments of offshore wind energy risk and resilience.

30) Section 5.2 - this section provides no new insight or results. Wind-wave misalignment is simply discussed but there is no actual evidence supporting the usefulness of the C-WFS framework. Consider adding such analysis (best option) or removing the section entirely (last resort since it is argued that wind-wave alignment is important).

This point has been addressed through our responses to Comments 27-29, and the related discussion now has been included to Section 4.4 in the revised manuscript.

31) L684 - a "more stable atmospheric boundary layer" was never actually shown. This was deduced from a lower surface enthalpy flux and lower TKE at a single level, but atmospheric stability is a function of height and typically shown through atmospheric profiles or the calculation of quantities such as the Richardson Number.

We thank the reviewer for this comment. The reviewer is correct that we inferred a "more stable atmospheric boundary layer" from indirect evidence (lower surface enthalpy flux and single-level TKE) rather than demonstrating it explicitly. As noted, atmospheric stability is fundamentally tied to the vertical structure of the boundary layer and is more appropriately assessed using the profiles or stability diagnostics (e.g., Richardson number) as suggested. In line with our earlier discussion, we have removed the relevant text from the revised manuscript. A more detailed examination of atmospheric stability will be addressed in a dedicated follow-up study.

32) L686-687 - while the values are slightly closer to observations, they are overall very far off which suggests that there is something inherently missing or wrong in the modeling system as a whole. Is it worth highlighting marginal increases in performance in something that is overall simulated very poorly?

This point has been addressed in our responses to Comments 24–26. Since the statement pertains to veer analysis, the related discussion has been removed from the revised manuscript.

33) L687-689 - these sentences are inaccurate and/or worded incorrectly. It reads as if saying ocean coupling isn't included in IEC standards. Additionally, it seems to make the claim that this is a novel finding of the paper.

Thank you for pointing this out. The IEC defines wave and wind conditions and acknowledges wind–wave directionality, but it is not quantified or mandated within the standard's load case definitions. In the revised manuscript, this statement has been removed in line with the deletion of the wind veer analysis.

34) L689-690 - are the comparisons to buoy data shown in the paper? How do we know that it's more accurate than the other two models when only the AOW experiment was shown?

The comparisons to buoy observations are presented in Figure 9 of the revised manuscript (Fig. C1 in the original manuscript). These show that the simulated wind and wave directions in the 'AOW' experiment reasonably match the observed values at the two buoy locations.

Regarding the second question, as discussed in our response to Comment 29, the 'A' and 'AO' can use prescribed wave parameters derived from external forcing datasets, which do not evolve interactively with the simulated storm. This leads to inconsistencies between wind and wave fields, particularly when the modeled storm deviates from the observed storm's track, intensity, or timing. In contrast, the fully coupled system allows for two-way interactions between the atmosphere and wave components, enabling the wind and wave fields to evolve consistently and in real time with the storms, thereby providing a more physically consistent and realistic representation. Additionally, to clarify, in our modeling setup, the 'A' and 'AO' simulations do not generate any wave-related parameters or variables.

35) L690-691 - after reading this paper, I'm not convinced of the critical risk of wind-wave misalignment. Additionally, both atmosphere-only and atmosphere-ocean models absolutely can simulate wind-wave misalignment through the boundary conditions.

As we have clarified in our previous responses (see Comments 27-29 and 34), we respectively emphasize a key distinction: atmosphere-only and atmosphere-ocean models without explicit wave coupling do not inherently simulate wind-wave misalignment. Instead, they can account for or represent wind-wave misalignment only if wave direction and other wave characteristics are prescribed as external boundary conditions. This approach has critical limitations. Please refer to our earlier responses for a detailed explanation.

36) L711-712 - if the 4 cases in the sea spray study are insufficient for generalizable results, then this study with a single case also must not be presented as generalizable. There is no "limitations" section in this paper although there are certainly many to be mentioned; this is one of them.

We appreciate the reviewer's point and want to clarify that we do not intend to generalize the results from this single-case study. Our primary goal is to introduce and demonstrate the capabilities of the newly developed atmosphere-ocean-wave coupled modeling system and to explore its potential to improve the representation of storm-scale wind structures relevant to offshore wind infrastructure risk assessment.

While the manuscript does not contain a dedicated "Limitations" section, several important limitations, including the single-case nature of the study, the absence of sea spray limitations, and unresolved sensitivities related to ocean model resolution, are explicitly discussed in Section 6 (Section 7 in the original manuscript). These discussions have been retained and slightly expanded in the revised manuscript for clarity.

**Technical Suggestions**

37) L124 - not a big deal, but the C-WFS acronym might be helped by underlining the letters for which it represents (coupled WRF-FVCOM-SWAN).

Completed.

38) L180-182 - similarly here: the "A" experiment wasn't immediately apparent, and I was thinking the next experiment would be "B" but it was "AO" and then I had to think a bit before registering "Atmosphere-Ocean." So it might be helpful to spell this out (e.g., "The WRF standalone simulation – representing the atmosphere only – is named experiment 'A'..."). It also may be worth considering aligning the experiment names with the acronym for the model framework (e.g., "WRF", "C-WF", "C-WFS") so that the "AOW" experiment is clearly the full C-WFS framework.

In the revised manuscript (Lines 139–142), we have explicitly defined the experiments 'A', 'AO', and 'AOW' upon first mention to improve clarity.

39) L326 - why is this figure mentioned in the paper but included in the appendix?

Figure A1 (now Figure A3) is placed in the appendix to streamline the main manuscript and maintain focus on the core findings. Several other figures are also moved to the appendix for similar reasons in the revised manuscript. While not shown in the main text, these appendix figures provide supporting evidence for the results discussed and are referenced accordingly to guide the reader to additional detail as needed.

40) Figure 6 - the observations are difficult to compare against the model. Would profiles not be better? The x-axis is also poorly formatted.

In the revised manuscript, this figure has been moved to the appendix (now Figure A2) to improve the flow of the main text. We have also revised the x-axis formatting for improved readability. Vertical profiles are provided in Figure 5 of the revised main manuscript.

41) L386-387 - "The area-averaged… Table 3" - this sentence has something wrong grammatically. Possibly at "calculated and OSTIA and"

The sentence and associated discussions are completely re-rewritten in the revised manuscript.

42) Fig 10a never referenced.

Figure 10a (now Figure 7a) is included to provide storm locations, so that readers do not need to refer back to Figure 3 for spatial orientation. To address this comment, we have now explicitly referenced Figure 7a in the revised manuscript to clarify its purpose.

43) L519 - missing word. Z0?

In the revised manuscript, this sentence has been completely re-written.

44) General comment: the section titles without punctuation are strange. For example, section 5 reads "5 Implication for Potential Risks…" as if there will be a list of 5 things.

We have followed the formatting guidelines provided by the Wind Energy Science (WES) journal, which recommends section headings without punctuation.

45) L560-563 - this refers to Fig 13, right?

Yes, this section originally referred to Figure 13. However, as previously discussed, the related discussion has been removed from the manuscript to maintain the primary focus and overall brevity.

46) L625 - why is this discussed below? Consider just discussing this here to give context to the reader.

We have re-written this section and incorporated it into Section 4.4 for brevity and clarity, ensuring that the relevant discussion is now presented in a more concise and focused manner.

**References**

Abdolali, A., Roland, A., van der Westhuysen, A., Meixner, J., Chawla, A., Hesser, T. J., Smith, J. M., & Sikiric, M. D.: Large-scale hurricane modeling using domain decomposition parallelization and implicit scheme implemented in WAVEWATCH III wave model. Coastal Engineering, 157. https://doi.org/10.1016/j.coastaleng.2020.103656, 2020.

Brown, A. R., Beljaars, A. C. M., Hersbach, H., Hollingsworth, A., Miller, M., & Vasiljevic, D.: Wind turning across the marine atmospheric boundary layer. Quarterly Journal of the Royal Meteorological Society, 131(612), 1233–1250. https://doi.org/10.1256/qj.04.163, 2005.

Cavaleri, L., Fox-Kemper, B., and Hemer, M.: Wind waves in the coupled climate system, Bull. Amer. Meteor. Soc., 93, 801 1651–1661, https://doi.org/10.1175/BAMS-D-11-00170.1, 2012.

Charnock, H., 1955: Wind stress on a water surface. Quart. J. Roy. Meteor. Soc., 81, 639–640.

Chen, X., Bryan, G. H., Zhang, J. A., Cione, J., and Marks, F. D.: A framework for simulating the tropical cyclone boundary layer using large-eddy simulation and its use in evaluating PBL parameterizations, Journal of Atmospheric Science, 78(11) 3559-3574, DOI: https://doi.org/10.1175/JAS-D-20-0227.1, 2021.

Chen, S. S., Zhao, W., Donelan, M. A., and Tolman, H. L.: Directional wind–wave coupling in fully coupled atmosphere–wave–ocean models: Results from CBLAST-Hurricane, *J. Atmos. Sci.*, 70, 3198–3215, 2013.

Creasey, R. L., and Elsberry, R. L.: Tropical cyclone center positions from sequences of HDSS sondes deployed along high-altitude overpasses. Wea. Forecasting, 32, 317–325, https://doi.org/10.1175/WAF-D-16-0096.1, 2017

Cummings, J. A. and Smedstad, O. M.: Ocean data impacts in global HYCOM, Journal of Atmospheric and Oceanic Technology, 31, 1771–1791, https://doi.org/10.1175/JTECH-D-14-00011.1, 2014.

Ding, J., Chen, H., Liu, X., Rashed, Y. F., and Fu, Z.: Fatigue reliability analysis of offshore wind turbines under combined wind-wave excitation via DPIM, *arXiv* v1, 14 pp., https://doi.org/10.48550/arXiv.2502.09429, 2025.

Donelan, M. A. et al. On the limiting aerodynamic roughness of the ocean in very strong winds. Geophys. Res. Lett. 31, L18306, 10.1029/2004GL019460, 2004.

Drennan, W. M., Taylor, P. K., and Yelland, M. J., "Parameterizing the sea surface roughness," J. Phys. Oceanogr. 35(5), 835–848, 2005.

Franklin, J. L., Black, M. L., and Valde, K.: GPS dropwindsonde wind profile in hurricanes and their operational implications. Weather and Forecasting, 18(1), 32-44, DOI: https://doi.org/10.1175/1520-0434(2003)018<0032:GDWPIH>2.0.CO;2, 2003.

Kapoor, A., Ouakka, S., Arwade, S. R., Lundquist, J. K., Lackner, M. A., Myers, A. T., Worsnop, R. P., and Bryan, G. H.: Hurricane eyewall winds and structural response of wind turbines, Wind Energy Science, 5(1), 89-104, https://doi.org/10.5194/wes-5-89-2020, 2020.

Kim, E. and Manuel, L.: Hurricane-induced loads on offshore wind turbines with considerations for nacelle yaw and blade pitch control, Wind Engineering, 38(4), https://doi.org/10.1260/0309-524X.38.4.413, 2014.

Oladejo, H. O., Bernstein, D. N., Cambazoglu, M. K., Nechaev, D., Abdolali, A., and Wiggert, J. D.: Wind forcing, source term and grid optimization for hurricane wave modelling in the Gulf of Mexico. Coastal Engineering, 197. https://doi.org/10.1016/j.coastaleng.2024.104692, 2025.

Powell, M. D., Vickery, P. J. and Reinhold, T. A. Reduced drag coefficient for high wind speeds in tropical cyclones. Nature 422, 279–283, 2003.

Pringle, W. J., and Kotamarthi, V. R.: Coupled Ocean Wave-Atmosphere Models for Offshore Wind Energy (Issue ANL/EVS-21/8). https://doi.org/10.2172/1829093, 2021.

Rose, S., Jaramillo, P., Small, M. J., and Apt, J.: Quantifying the hurricane risk to offshore wind turbines, Proc. Natl. Acad. Sci. USA, 109, 3247–3252, https://doi.org/10.1073/pnas.1111769109, 2012.

Sanchez Gomez, M., Lundquist, J. K., Mirocha, J. D., and Arthur, R. S.: Investigating the physical mechanisms that modify wind plant blockage in stable boundary layers, Wind Energy Science, 8, 1049–1069, https://doi.org/10.5194/wes-8-1049-2023, 2023.

Shanahan, T. and Fitzgerald, B.: Wind–Wave Misalignment in Irish Waters and Its Impact on Floating Offshore Wind Turbines. *Energies*, *18*(2), 372; https://doi.org/10.3390/en18020372, 2025.

Shimura, T., Noh, Y., and Hara, T.: Long-term impacts of ocean wave-dependent roughness on global climate systems, J. Geophy. Reasrch: Oceans, 122(3), 1995-2011, https://doi.org/10.1002/2016JC012621, 2017.

Taylor, P. K. and Yelland, M. J., "The dependence of sea surface roughness on the height and steepness of the waves," J. Phys. Oceanogr. 31(2), 572–590, 2001.

Tolman, H. L., Alves, J-H G. M., and Chao, Y. Y.: Operational forecasting of wind-generated waves by Hurricane Isabel at NCEP, Weather and Forecasting, 20(4), 544–557, DOI: https://doi.org/10.1175/WAF852.1, 2006.

Verma, A.S., Jiang, Z., Ren, Z. et al. Effects of Wind-Wave Misalignment on a Wind Turbine Blade Mating Process: Impact Velocities, Blade Root Damages and Structural Safety Assessment. J. Marine. Sci. Appl. 19, 218–233 (2020). https://doi.org/10.1007/s11804-020-00141-7

Xu, X., Voermans, J. J., Zhang, W., Zhao, B., Qiao, F., Liu, Q., Moon, I.-J., Janekovic, I., Waseda, T., and Babanin, A. V.: Tropical cyclone modeling with the inclusion of wave-coupled processes: sea spray and wave turbulence, Geophys. Res. Lett., 50, e2023GL106536, https://doi.org/10.1029/2023GL106536, 2023.

Wada, A. and Usui, N.: Impacts of oceanic preexisting conditions on predictions of Typhoon Hai-Tang in 2005, Advances in Meteorology, 2010, 756071, 2010.

Warner, J. C., Armstrong, B., He, R., and Zambon, J. B.: Development of a coupled ocean–atmosphere–wave–sediment transport (COAWST) modeling system, Ocean Model., 35, 230–244, https://doi.org/10.1016/j.ocemod.2010.07.010, 2010.

Worsnop, R. P., Lundquist, J. K., Bryan, G. H., Damiani, R., and Musial, W.: Gusts and shear within hurricane eyewalls can exceed offshore wind turbine design standards, Geophys. Res. Lett., 44, 6413–6420, doi:10.1002/ 2017GL073537, 2017.

Zambon, J. B., He, R., and Warner, J. C.: Investigation of Hurricane Ivan using the coupled ocean–atmosphere–wave–sediment transport (COAWST) model, Ocean Dynamics, 64, 1535–1554, https://doi.org/10.1007/s10236-014-0777-7, 2014.

Zhang, F., Pu, Z., and Wang, C.: Effects of boundary layer vertical mixing on the evolution of hurricanes over land. Monthly Weather Review, 145(6), 2343–2361. https://doi.org/10.1175/MWR-D-16-0421.1, 2017.

---

## Author Comment (AC2)

Review of "Fully Coupled High-Resolution Atmosphere-Ocean-Wave Simulations of Hurricane Henri (2021): Implications for Offshore Load Assessments"

This study investigates the effects of using different combinations of COAWST modeling components on the simulated Hurricane structure: Atmosphere only, Atmosphere-Ocean, and Atmosphere-Ocean-Wave. Various measurements are used to validate the results, and help to interpret the mechanisms of the interactions between atmosphere, wave and ocean. It also discusses the relevance to the calculation of loads for offshore wind turbines.

The paper is well written, with clear research questions, reasonable methodology, convincing presentation of results and analysis. I recommend the paper Minor Revision, with the following comments and suggestions.

We are sincerely grateful to the reviewer for their time, thoughtful comments, and constructive questions, all of which have significantly enhanced the clarity and robustness of our manuscript. Below, we provide a detailed, line-by-line response to each of the reviewer's comments.

In addition, we have substantially reduced the manuscript's length while preserving the core ideas and analyses, in line with Reviewer #1's suggestions. Our revisions emphasize the primary objectives of the study: introducing and demonstrating a new atmosphere-ocean-wave coupled modeling system, and examining how wave dynamics influence TC evolution, particularly wind structure, which is a critical factor in risk assessment and the design of offshore wind energy infrastructure.

To maintain a focused and concise scope, we have reformatted our introduction to describe our motivation for the study more clearly and removed the original implications section from the revised manuscript. The wind-wave misalignment and alignment analysis has been incorporated in Section 4.4: Ocean Surface Waves. The wind veer analyses have been omitted, and a more comprehensive wind energy-focused analysis, leveraging the fully coupled model introduced in this study, will be presented in a dedicated follow-up study.

Sec 2.1, line 132: it is written that "we have modified the WRF code to enable…". How does this work with the SWAN model? Isn't the Taylor-Yelland roughness length algorithms included in SWAN?

Thank you for this important question. SWAN calculates and provides wave parameters such as significant wave height and steepness; however, it does not directly compute surface roughness length ($z_0$). Instead, $z_0$ is derived from coupled atmospheric–wave process, as it depends on both wave characteristics and wind stress, which are parameterized within the atmospheric component (WRF).

To ensure that wave effects are reflected in the surface roughness seen by WRF, we modified WRF surface layer schemes to ingest wave parameters, specifically significant wave height and peak wavelength, from SWAN. These parameters are used in WRF's surface scheme to compute the surface roughness length following the Taylor and Yelland (2001) formulation. This allows the atmosphere model to account for sea state-dependent surface drag based on dynamically evolving wave conditions, rather than relying on default roughness parameterizations that do not consider wave information.

Sec 3.1, line 198: "a resolution of 4 km or less is adequately convection-permitting in WRF for simulating extreme events" – what do you mean "adequate"?

We have revised the sentence to: "4 km resolution or less supports convection-permitting simulations" (lines 138–140). This resolution range has been widely used in previous studies to simulate extreme weather events such as tropical cyclones (TCs) and mesoscale convective systems, where it has been shown to capture key convective features, storm structures, and precipitation patterns more realistically than coarser-resolution models with parameterized convection (e.g., Gentry and Lackmann, 2010; Prein et al., 2015).

The paragraph starting from line 200 seems to be about the ocean modeling. Yet the last sentence seems to belong to the atmospheric modeling, or does it?

The section has been completely restructured to clearly distinguish the atmospheric and ocean modeling components in Section 3.1.

Fig 3: many figures, including this one, miss x and y-axes labels. Please add all of them.

Thank you for pointing this out. We have added descriptive labels to both the x- and y-axes in Figure 3, as well as in all other figures (Figures 2, 4, 7, 8, 9, 10, 11, 12, and S1), where they were previously missing, to improve clarity.

Fig 3: What is the temporal resolution of the Best Track data in Fig 3b? What are the heights of winds from the modeling in Fig 3c?

We used the International Best Track Archive for Climate Stewardship (IBTrACS) for the best track data shown in Figs. 3a-c. The dataset provides storm information at 6 hourly temporal resolution. For Fig. 3c, the winds from the model are 10-meter winds derived from WRF output, consistent with standard practices for surface wind analysis. We have clarified both the temporal resolution of the best track data and the wind height from the model in the revised figure caption and corresponding text in lines 156-158.

Fig 4: Did the authors explain why the two time are chosen for the analysis? What about other time? Please add Unit to the color bars – not only to this figure but to all figures.

We have selected these two times because NOAA WP-3D airborne Doppler radar (TC-RADAR) observations are only available at those specific times, providing high-resolution three-dimensional wind structure data from 500 m to 9 km altitude. These observations offer a valuable reference for evaluating model performance, especially given that such detailed airborne radar measurements are not routinely available for every TC. The availability of TC-RADAR data during Hurricane Henri represents a unique opportunity to conduct a more rigorous assessment of the modeled wind field. While we examined model outputs at multiple times as part of our internal analysis, we found that the spatial distribution and structure were generally consistent with the times shown.

Additionally, we have added units to the color bars in Fig. 4, as well as to all other relevant figures throughout the manuscript as suggested.

Fig 7: The profiles in this figure, is it an average over a certain period or is it a snapshot on the day?

The profiles in this figure (now Fig. 6 in the revised manuscript) are based on dropsondes that were released during a single NOAA WP-3D flight that traversed the storm center from east to west within approximately 50 minutes from 23:21 UTC on 21 to 00:11 UTC on 22 August 2020 (with exact times indicated in Fig 3d). For the model results, we used a single output time at 00:00 UTC on 22 August 2021 to align with the timing of the observational data. We have clarified this in the revised main text (lines 166-168) and in the captions for Figs. 3 and 6.

Also, these dropsondes present azimuthally averaged vertical wind profiles in the inner-eyewall ($0.2 \leq$ r/RMW $\leq 1$) and outer-eyewall ($2 \leq$ r/RMW $\leq 2.5$) regions, based on the locations of the seven dropsondes highlighted in blue dots in Fig. 3d.

Fig 10a: add legend. Fig 10e and f: do you think the difference between wind and wave directions is caused by swell? What is the model's ability in capturing swell?

Thank you for pointing this out. We have added appropriate legends to Fig. 10 (now Figure 8 in the revised manuscript) in the revised manuscript to improve clarity.

The wave model should be able to capture swell adequately, but we do not believe this is the reason for the differences. Previous studies (e.g., Tolman et al., 2005) have shown that wave accuracy is highly sensitive to the quality of atmospheric forcing, particularly during hurricanes. This sensitivity helps explain the discrepancies observed in our wave model results shown in Fig. 10 (Fig. 8 in the revised manuscript). We attribute these differences primarily to wind forcing and the relative position of each buoy to the TC center. For instance, the overprediction of wind speed at buoy 41001 around 00 UTC on 22 August leads to an overestimation of wave height. Additionally, because the simulated TC track passes farther east than the observed track when approaching buoy 41002, the discrepancy in the timing of the wave direction shift can be attributed to this track error.

Please make sure all figures have axes labels, legends and color bars with units.

Thank you for the observation. We have carefully reviewed all figures in the manuscript and ensured that each includes clearly labeled axes, appropriate legends, and color bars with units where applicable. These updates have been made throughout the revised manuscript to improve clarity and consistency.

**Citation**: https://doi.org/10.5194/wes-2025-47-RC2

**References**

Gentry, M. A. and Lackmann, G. M.: Sensitivity of simulated tropical cyclone structure and intensity to horizontal resolution, Monthly Weather Review, 138, 688–704, 2010.

Prein, A. F., Langhans, W., Fosser, G., Ferrone, A., Ban, N., Goergen, K., Keller, M., Tölle, M., Gutjahr, O., Feser, F., et al.: A review on regional convection-permitting climate modeling: Demonstrations, prospects, and challenges, Reviews of Geophysics, 53, 323–361, https://doi.org/10.1002/2014RG000475, 2015.

Tolman, H.L., Alves, J-H G. M., and Chao, Y. Y.: Operational forecasting of wind-generated waves by Hurricane Isabel at NCEP, Weather and Forecasting, 20(4), 544-557, DOI: https://doi.org/10.1175/WAF852.1, 2006.